# `Framer`: Interactive Frame Interpolation

**Wen Wang**[1,2], **Qiuyu Wang**[2], **Kecheng Zheng**[2], **Hao Ouyang**[2], **Zhekai Chen**[1],
**Biao Gong**[2], **Hao Chen**[1], **Yujun Shen**[2], **Chunhua Shen**[3,1,2]

[1] Zhejiang University  [2] Ant Group  [3] Zhejiang University of Technology

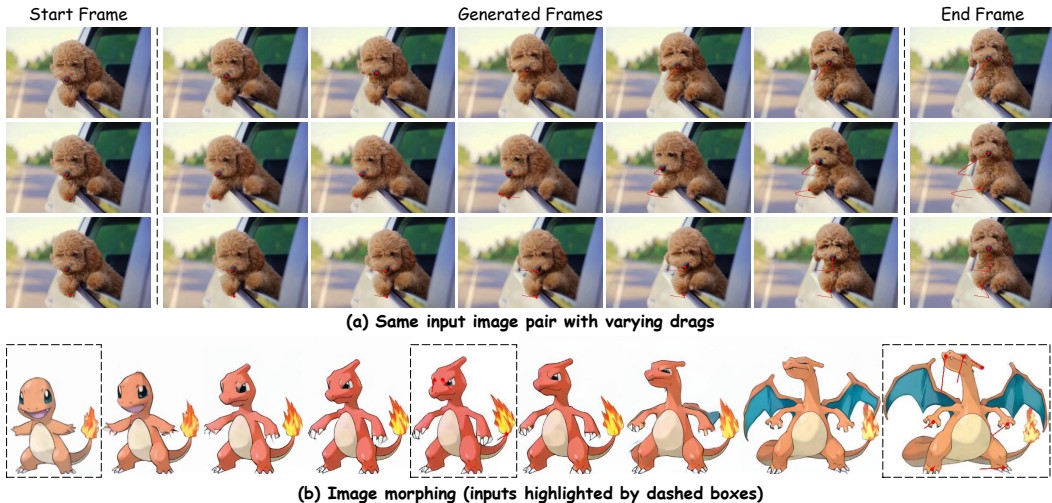

Figure 1: Showcases produced by our `Framer`. It facilitates fine-grained customization of local motions and generates varying interpolation results given the same input start and end frame pair (first 3 rows). Moreover, `Framer` handles challenging cases and can realize smooth image morphing (last 2 rows). The input trajectories are overlayed on the frames.

## Abstract

We propose `Framer` for interactive frame interpolation, which targets producing smoothly transitioning frames between two images as per user creativity. Concretely, besides taking the start and end frames as inputs, our approach supports customizing the transition process by tailoring the trajectory of some selected keypoints. Such a design enjoys two clear benefits. First, incorporating human interaction mitigates the issue arising from numerous possibilities of transforming one image to another, and in turn enables finer control of local motions. Second, as the most basic form of interaction, keypoints help establish the correspondence across frames, enhancing the model to handle challenging cases (*e.g.*, objects on the start and end frames are of different shapes and styles). It is noteworthy that our system also offers an "autopilot" mode, where we introduce a module to estimate the keypoints and refine the trajectory automatically, to simplify the usage in practice. Extensive experimental results demonstrate the appealing performance of `Framer` on various applications, such as image morphing, time-lapse video generation, cartoon interpolation, *etc.* The code, model, and interface are publicly accessible at https://github.com/aim-uofa/Framer.

## 1 Introduction

The creation of seamless and visually appealing transitions between frames (Dong et al., 2023) is a fundamental requirement in various applications, including image morphing (Aloraibi, 2023), slow-motion video generation (Reda et al., 2022), and cartoon interpolation (Xing et al., 2024). Users

often need to control the motion trajectories, deformation dynamics, and temporal coherence of interpolated frames to achieve specific outcomes. Therefore, incorporating interactive capabilities into frame interpolation frameworks is crucial for expanding the practical applicability.

Traditional video frame interpolation methods (Jiang et al., 2018; Xu et al., 2019; Liu et al., 2020; Niklaus & Liu, 2020; Sim et al., 2021; Lee et al., 2020; Ding et al., 2021) often rely on estimating optical flow or motion to predict intermediate frames deterministically. While significant progress has been made in this area, these approaches struggle in scenarios involving large motion or substantial changes in object appearance, due to an inaccurate flow estimation. What's more, when transforming one image to another, there can be numerous plausible ways objects and scenes can transition. A deterministic result may not align with user expectations or creative intent.

Orthogonal to existing methods, we propose `Framer`, an interactive frame interpolation framework designed to produce smoothly transitioning frames between two images. Our approach allows users to customize the transition process by tailoring the trajectories of selected keypoints, thus directly influencing the motion and deformation of objects within the scene. Such design offers two significant benefits. **First**, the incorporation of keypoint-based interaction resolves the ambiguity inherent in transforming one image into another, allowing for precise control over how specific regions of the image move and change. As shown in Fig. 1a, users can control the movements of the dog's paw and head through simple and intuitive interactions. **Second**, keypoint trajectories establish explicit correspondences across frames, which is especially beneficial in challenging cases where objects change in shape, style, or even semantic meaning. As shown in Fig. 1b, the keypoint trajectories establish the correspondences between keypoints from Pokémon in varying forms and help produce a smooth "evolution" process of Pokémon.

Concretely, we view video frame interpolation from a generative perspective and finetune a large-scale pre-trained image-to-video diffusion model (Blattmann et al., 2023a) on open-domain video datasets (Nan et al., 2024) to facilitate video frame interpolation. The additional last-frame conditioning is introduced during the fine-tuning process. Afterward, a point trajectory controlling branch is introduced to take the additional point trajectory inputs, thus guiding the video interpolation process. During inference, `Framer` supports the "interactive" mode for customized video frame interpolation, following user-input point trajectories.

Understanding that manual keypoint annotation may not always be desirable, we offer an "autopilot" mode for `Framer`. Technically, we propose a novel bi-directional point-tracking method that estimates the trajectories of matched points over the entire video sequence, by analyzing both forward and backward motions between frames. It automates the process of obtaining keypoint trajectories, enabling `Framer` to generate motion-natural and temporally coherent interpolation results without requiring extensive user input. The "autopilot" mode simplifies the workflow while still benefiting from the enhanced correspondence provided by the points trajectories.

We conduct extensive experiments to evaluate the performance of `Framer` across various applications, including image morphing, time-lapse video generation, and cartoon interpolation. The results demonstrate that `Framer` produces smooth and visually appealing transitions, outperforming existing methods, particularly in cases involving complex motions and significant appearance changes. By combining the strengths of generative models with user-guided interactions, `Framer` improves both the quality and controllability of the interpolated frames.

## 2 RELATED WORK

### 2.1 VIDEO FRAME INTERPOLATION

Video frame interpolation (VFI) aims to synthesize intermediate frames from two successive video frames. Most previous methods view VFI as a low-level task, assuming a moderate motion between frames. These methods can roughly be categorized as flow-based methods and kernel-based methods. Specifically, the flow-based methods leverage estimated optical flow for frame synthesis (Jiang et al., 2018; Xu et al., 2019; Liu et al., 2020; Niklaus & Liu, 2020; 2018; Sim et al., 2021; Huang et al., 2020; Jin et al., 2023; Xue et al., 2019; Park et al., 2020; 2021; Kong et al., 2022; Yu et al., 2021). By contrast, the kernel-based methods rely on spatially adaptive kernels to synthesize the interpolated pixels (Lee et al., 2020; Cheng & Chen, 2022; Ding et al., 2021; Niklaus

et al., 2017; Cheng & Chen, 2020; Gui et al., 2020; Lu et al., 2022). While the former potentially suffers from inaccurate flow estimation, the latter are often constrained by kernel size. To obtain the best of both worlds, some methods combine the flow- and kernel-based methods for end-to-end video frame interpolation (Bao et al., 2019; 2021; Danier et al., 2022; Li et al., 2022). Realizing that motion ambiguity remains given the start and end frames (Zhou et al., 2023; Zhong et al., 2024), Zhong et al. (2024) resolves the ambiguities by providing explicit hints on how far objects have traveled between start and end frames, termed "distance indexing". While Zhong et al. (2024) supports user-interaction, the distance-indexing interaction requires setting detailed distance values for middle frames, making them less intuitive.

Recently, inspired by the generative capacity of large-scale pre-trained video diffusion models, some methods attempt to tackle VFI from a generation perspective (Danier et al., 2024; Feng et al., 2024; Jain et al., 2024; Xing et al., 2023; Wang et al., 2024a). For example, LDMVFI (Danier et al., 2024) formulates VFI as a conditional generation problem and utilizes a latent diffusion model for perceptually oriented video frame interpolation. Similarly, VIDIM (Jain et al., 2024) leverages cascaded diffusion models to generate high-fidelity interpolated videos with nonlinear motions. Though progress has been made, these methods still have difficulties in tackling large differences between input frames. Moreover, they generate a single deterministic solution for video frame interpolation, without controllability. Differently, we can generate multiple plausible solutions under large motion changes, and allow simple and intuitive drag interaction for user-intended results.

Video frame interpolation has a wide range of applications in many fields. While traditional interpolation methods focus on improving the frame rate of the input video (Li et al., 2023; Huang et al., 2020; Kalluri et al., 2023; Reda et al., 2022), generative frame interpolation methods take advantage of large-scale pre-trained diffusion models, and are more concerned with dealing with situations where the input frames have large differences (Xing et al., 2023; Wang et al., 2024a). In addition, some works train tailored video frame interpolation models for specific application scenarios, such as cartoon interpolation (Siyao et al., 2021; Chen & Zwicker, 2022; Xing et al., 2024), sketch interpolation (Siyao et al., 2023; Shen et al., 2024), etc. In this paper, we show that `Framer` can handle all of the above tasks under a unified framework, and allow users to achieve fine-grained control of the interpolation process through simple interactions.

## 2.2 Video Diffusion Models

Large-scale pre-trained video diffusion models (Brooks et al., 2024; Blattmann et al., 2023b; Chen et al., 2024; Blattmann et al., 2023a) have shown unprecedented generation results in visual quality, diversity, and realism. These methods leverage text or starting image controls, which are often insufficient in precision. Inspired by the success in controllable image generation (Zhang et al., 2023b; Mou et al., 2024b), several works attempt to add additional controls to video diffusion models. Early explorations (Wang et al., 2023; Guo et al., 2023) utilize structural controls, like sketch and depth maps, for video generation. However, these control signals are difficult to obtain during sampling, limiting their practical applications. Differently, recent works focus on motion control and introduce trajectory control for object motion (Wu et al., 2024; Mou et al., 2024a; Yin et al., 2023) and camera pose control for camera motion (Wang et al., 2024b; He et al., 2024; Bahmani et al., 2024). In this paper, we enhance the creative potential and flexibility of the video frame interpolation process, allowing users to produce plausible results following their control.

## 3 Method

Given two frames, $I^0$ and $I^n$, indicating the start and end frame in a video, our goal is to generate the plausible contiguous video $I = \{I^i\}_{i=0}^n$ by sampling from the conditional distribution $p\left(I \mid I^0, I^n\right)$. Here, $n$ is the number of frames in the video. Our method, termed `Framer`, supports a user-interactive mode for customized point trajectories and an "autopilot" mode for video frame interpolation without trajectory inputs, as shown in Fig. 2a and Fig. 2b. In the following, we will introduce how we add frame conditions to the video diffusion model to achieve video interpolation in Sec. 3.1. To support user-interactive drag control, we introduce a control branch in Sec. 3.2 for point trajectory guidance, which also enhances point correspondences across frames. In the "autopilot" mode, we estimate trajectories of matched points in the video with our novel bi-directional point tracking method, as illustrated in Sec. 3.3.

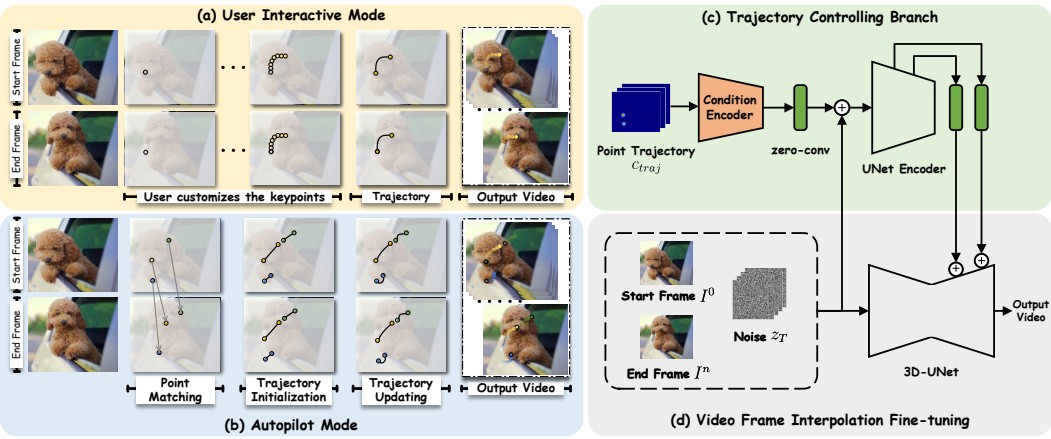

Figure 2: Framer supports (a) a user-interactive mode for customized point trajectories and (b) an "autopilot" mode for frame interpolation without trajectory inputs. During training, (d) we fine-tune the 3D-UNet of a pre-trained video diffusion model for frame interpolation. Afterward, (c) we introduce point trajectory control by freezing the 3D-UNet and fine-tuning the controlling branch.

## 3.1 MODEL ARCHITECTURE

Large-scale pre-trained video diffusion models have a strong visual prior on the appearance, structure, and movement of open-world objects (Brooks et al., 2024). Our approach builds on the video diffusion model to exploit this prior. Considering that the Image-to-Video (I2V) diffusion model naturally supports first-frame conditioning, we choose the representative I2V diffusion model, Stable Video Diffusion (SVD) (Blattmann et al., 2023a), as our base model, as shown in Fig. 2d.

Based on the I2V model, we need to introduce additional end-frame conditioning to realize video interpolation. To preserve the visual prior of the pre-trained SVD as much as possible, we follow the conditioning paradigm of SVD and inject end-frame conditions in the latent space and semantic space, respectively. Specifically, we concatenate the VAE-encoded latent feature of the first frame, denoted as $z^0$, with the noisy latent of the first frame, as did in SVD. Additionally, we concatenate the latent feature of the last frame, $z^n$, with the noisy latent of the end frame, $z_t^n$, considering that $z_t^n$ is derived by adding noise to $z^n$. In addition, we extract the CLIP image embedding of the first and last frames separately and concatenate them for cross-attention feature injection. The U-Net $\epsilon_\theta$ is trained using the denoising score matching objective:

$$\mathcal{L} = \mathbb{E}_{z_t, z^0, z^n, t, \epsilon \sim \mathcal{N}(0, \mathbf{I})} \left[ \left\| \epsilon - \epsilon_\theta \left( z_t; t, z^0, z^n \right) \right\|^2 \right]. \tag{1}$$

## 3.2 INTERACTIVE FRAME INTERPOLATION

Ambiguity remains given the start and end frames, especially when the distinction between the two frames is large. The reason is that multiple plausible interpolation results can be obtained by sampling video from the conditional distribution $P\left(I \mid I^0, I^n\right)$ for the same input pair. To better align with the user intention, we introduce a control branch for custmized point trajectory guidance.

Technically, we train a point trajectory-based control branch for correspondence modeling, as shown in Fig. 2c. During training, we use the following steps to obtain the point trajectory as control signals. Firstly, we randomly initialize some sampled points around a fixed sparse grid in the first frame, and use Co-Tracker (Karaev et al., 2023) to obtain the trajectories of these points in the whole video. Secondly, we remove trajectories that are not visible in more than half of the video frames. Lastly, we sample the point trajectories with larger motions with greater probability. Considering that the users usually only input a small number of point trajectories, we keep only 1 to 10 trajectories during training. Please refer to the Appendix A for more details.

After obtaining the sampled point trajectories, we follow DragNUWA (Yin et al., 2023) and DragAnything (Wu et al., 2024) to transform the point coordinates into a Gaussian heatmap, denoted as $c_{traj}$, which is used as input to the control module. We follow the conditioning mechanism in

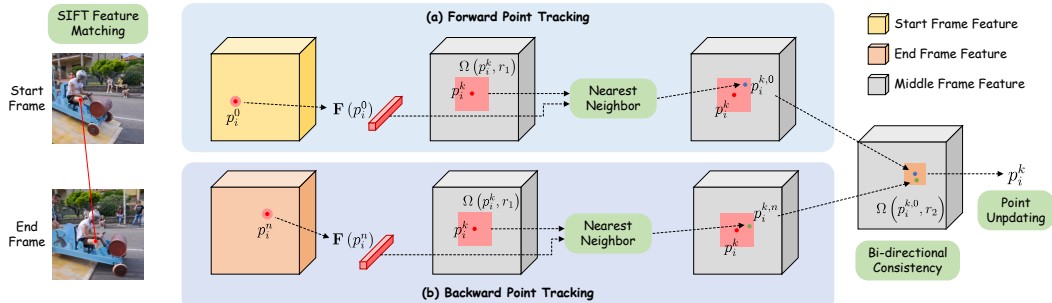

Figure 3: **Point trajectory estimation.** The point trajectory is initialized by interpolating the coordinates of matched keypoints. In each de-noising step, we perform point tracking by finding the nearest neighbor of keypoints in the start and end frames, respectively. Lastly, We check the bi-directional tracking consistency before updating the point coordinate.

ControlNet (Zhang et al., 2023b) to incorporate the trajectory control. Specifically, we copy the encoder of 3D-UNet to encode the trajectory map and add it into the decoder of U-Net after zero-convolution (Zhang et al., 2023b). This training process can be represented as:

$$\mathcal{L} = \mathbb{E}_{z_t, z^0, z^n, t, \epsilon \sim \mathcal{N}(0, \mathbf{I})} \left[ \left\| \epsilon - \epsilon_\theta^c \left( z_t; t, z^0, z^n, c_{traj} \right) \right\|^2 \right]. \tag{2}$$

Here, $\epsilon_\theta^c$ is the combination of the denoising U-Net and the ControlNet branch.

**Discussion.** The introduction of point trajectory control not only facilitates user interaction, but also enhances the correspondence among points from different frames. As demonstrated in experiments, this approach enables the model to effectively tackle challenging cases, such as when the start and end frames differ significantly.

### 3.3 "AUTOPILOT" MODE FOR FRAME INTERPOLATION

In practical applications, users may not always prefer manual drag controls. For this reason, we propose an "autopilot" mode to enhance the ease of use of our `Framer`. It mainly contains a trajectory initialization and a trajectory updating process, as illustrated in Fig. 2b.

**Trajectory Initialization.** Given the start and end frames of the input video, we can obtain the matching points between the two frames by applying feature-matching algorithms. The matched points are denoted as $\{\boldsymbol{p}_i\}_{i=1}^m$, where $m$ is the number of matching points. $\boldsymbol{p}_i$ denotes the known anchor points on the trajectory. At initialization, it contains the matched points on the first and last frames, *i.e.*, $\boldsymbol{p}_i = [p_i^0, p_i^n]$. Although varying feature matching algorithms are feasible, we use the classical SIFT feature matching (Lowe, 2004) here for its simplicity and effectiveness. Subsequently, we can obtain the $i$-th trajectory $\hat{c}_i$ by interpolating the anchor points $\boldsymbol{p}_i$. The estimated trajectory for all $m$ matched points, denoted as $\hat{c}_{traj} = \{\hat{c}_i\}_{i=1}^m$, are used as the input condition in Eq. (2).

**Trajectory Updating.** Although the initial trajectory provides temporally consistent point correspondence, the trajectory obtained by connecting points in the first and last frames may not be accurate. Inspired by DragGAN (Pan et al., 2023) and DragDiffusion (Shi et al., 2023), we perform point tracking using the intermediate feature in U-Net to update the trajectories. Specifically, in each denoising step, we interpolate the U-Net features to the image resolution, denoted as $\mathbf{F}$. Here we use the feature of the penultimate upsampled block in U-Net, since it enjoys a good trade-off between feature resolution and discriminativeness. We use $\mathbf{F}(p)$ to represent the feature of the point $p$, which is obtained via bilinear interpolation, since the coordinates may not be integers.

In each denoising step, we apply point tracking to update the coordinates of the middle frame points. We use nearest neighbor search in a feature patch around the point. The feature patch represents a set of points whose distance to point $p$ is less than $r$, and is denoted as $\Omega(p, r) = \{(x, y) \mid |x - x_p| < r, |y - y_p| < r\}$. For a middle frame point $p_i^k$ in the $k$-th frame, we find the

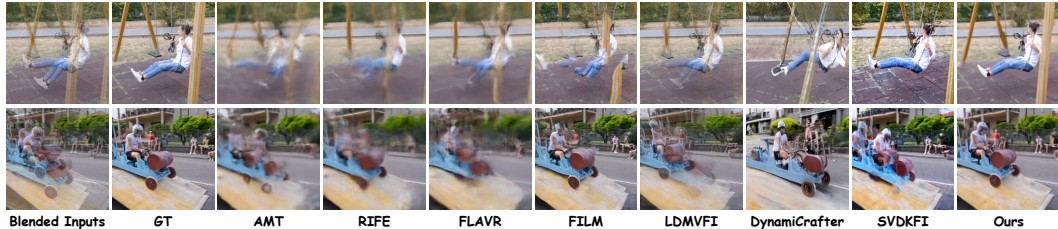

Figure 4: Qualitative comparison. For each method, we only present the middle frame of 7 interpolated frames. The full results can be seen in Fig. S7 and Fig. S8 in the Appendix.

nearest point relative to the anchor point $p_i^0$ via:

$$p_i^{k,0} := \underset{q_i^k \in \Omega\left(p_i^k, r_1\right)}{\arg\min} \left\| \mathbf{F}\left(q_i^k\right) - \mathbf{F}\left(p_i^0\right) \right\|_1. \qquad (3)$$

Similarly, we can obtain the nearest point relative to the last anchor point $p_i^n$:

$$p_i^{k,n} := \underset{q_i^k \in \Omega\left(p_i^k, r_1\right)}{\arg\min} \left\| \mathbf{F}\left(q_i^k\right) - \mathbf{F}\left(p_i^n\right) \right\|_1. \qquad (4)$$

As shown in Fig. 3, to further ensure the accuracy of the coordinates of the updated points, we check the consistency of the two nearest points obtained by matching with $p_i^0$ and $p_i^n$. When the distance between the two is less than a threshold $r_2$, i.e., $p_i^{k,n} \in \Omega\left(p_i^{k,0}, r_2\right)$, we update the point coordinates by setting $p_i^k = (p_i^{k,0} + p_i^{k,n})/2$. Then, we add the point to the anchor points list $\boldsymbol{p}_i$ and interpolate $\boldsymbol{p}_i$ to get the updated trajectory $c_i$, which is used as the input condition to the next denoising step. The point trajectory estimation process is also illustrated in Alg. 1 in the Appendix.

## 4 EXPERIMENTS

### 4.1 IMPLEMENTATION DETAILS

Our method is built on SVD and trained on the high-quality OpenVidHD-0.4M dataset (Nan et al., 2024). The model is trained in two stages. Specifically, we first fine-tune the U-Net to accept the end frame conditions. Then, we train the controlling branch for point trajectory guidance. During the training of U-Net, we fixed the spatial attention and residual blocks, and only fine-tuned the input convolutional and temporal attention layers. The model is trained for 100K iterations using the AdamW optimizer Loshchilov & Hutter (2019) with a learning rate of $1e{-}5$. When training the control module, we fixed the U-Net and optimized the control module for 10K steps using the AdamW optimizer, with a learning rate of $1e{-}5$. We obtained the point trajectories by pre-processing the video using the Co-Tracker (Karaev et al., 2023). All training is performed on 16 NVIDIA A100 GPUs, and the total batch size is 16. The training takes about 2 days. During sampling, it takes about 4.64 seconds to generate 7 interpolated frames on the DAVIS-7 dataset. On average, it takes 0.67 seconds to produce a single interpolated frame. During "autopilot" mode sampling, we keep $m = 5$ best matching keypoints for trajectory guidance, and the distance thresholds for point tracking are set as $r_1 = 5$ and $r_2 = 3$. Please refer to Appendix A for more details.

### 4.2 COMPARISON

Existing methods do not support drag-user interaction. Thus, we use the "autopilot" mode of `Framer` to make fair comparisons. We select baselines from two distinct categories. The first category includes the latest general diffusion-based video interpolation models, including LDMVFI (Danier et al., 2024), DynamicCrafter (Xing et al., 2023), and SVDKFI (Wang et al., 2024a). The second category encompasses traditional video interpolation methods, such as AMT (Li et al., 2023), RIFE (Huang et al., 2020), FLAVR (Kalluri et al., 2023), and FILM (Reda et al., 2022). We conduct

|  | **DAVIS-7** | | | | | **UCF101-7** | | | | | |
|---|---|---|---|---|---|---|---|---|---|---|---|
|  | PSNR↑ | SSIM↑ | LPIPS↓ | FID↓ | FVD↓ | PSNR↑ | SSIM↑ | LPIPS↓ | FID↓ | FVD↓ | Latency |
| AMT (Li et al., 2023) | 21.66 | **0.7229** | 0.2860 | 39.17 | 245.25 | 26.64 | 0.9000 | 0.1878 | 37.80 | 270.98 | 0.165 |
| RIFE (Huang et al., 2020) | **22.00** | 0.7216 | 0.2663 | 39.16 | 319.79 | **27.04** | **0.9020** | 0.1575 | 27.96 | 300.40 | 0.072 |
| FLAVR (Kalluri et al., 2023) | 20.94 | 0.6880 | 0.3305 | 52.23 | 296.37 | 26.50 | 0.8982 | 0.1836 | 37.79 | 279.58 | 0.028 |
| FILM (Reda et al., 2022) | 21.67 | 0.7121 | **0.2191** | **17.20** | 162.86 | 26.74 | 0.8983 | **0.1378** | **16.22** | 239.48 | 0.291 |
| LDMVFI (Danier et al., 2024) | 21.11 | 0.6900 | 0.2535 | 21.96 | 269.72 | 26.68 | 0.8955 | 0.1446 | 17.55 | 270.33 | 9.340 |
| DynamiCrafter (Xing et al., 2023) | 15.48 | 0.4668 | 0.4628 | 35.95 | 468.78 | 17.62 | 0.7082 | 0.3361 | 61.71 | 646.91 | 13.166 |
| SVDKFI (Wang et al., 2024a) | 16.71 | 0.5274 | 0.3440 | 26.59 | 382.19 | 21.04 | 0.7991 | 0.2146 | 44.81 | 301.33 | 42.923 |
| Framer (Ours) | 21.23 | 0.7218 | 0.2525 | 27.13 | **115.65** | 25.04 | 0.8806 | 0.1714 | 31.69 | **181.55** | 4.644 |
| Framer with Co-Tracker (Ours) | 22.75 | 0.7931 | 0.2199 | 27.43 | 102.31 | 27.08 | 0.9024 | 0.1714 | 32.37 | 159.87 | 4.644 |

Table 1: Quantitative comparison with existing video interpolation methods on reconstruction and generative metrics, evaluated on all 7 generated frames. The latency for generating 7 intermediate frames is assessed on the NVIDIA A6000 GPU, using seconds as the measurement metric.

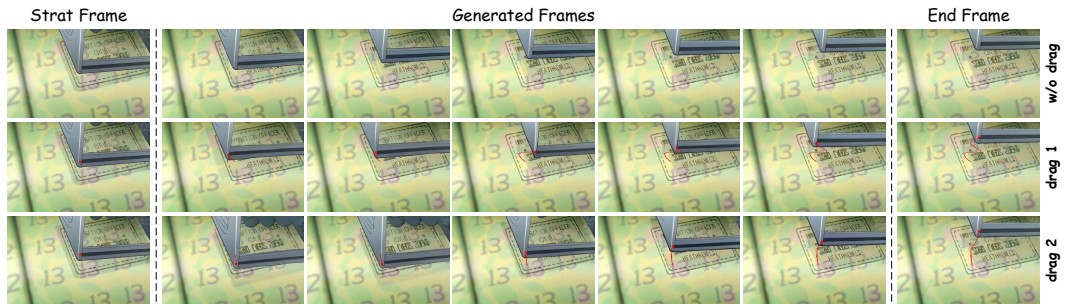

Figure 6: Results on user interaction. The first row is generated without drag input, while the other two are generated with different drag controls. Customized trajectories ares overlaid on frames.

quantitative and qualitative analyses, as well as user studies, on two publicly available datasets: DAVIS (Pont-Tuset et al., 2017) and UCF101 (Soomro et al., 2012).

**Qualitative Comparison.** As shown in Fig. 4, our method produces significantly clearer textures and natural motion compared to existing interpolation techniques. It performs especially well in scenarios with substantial differences between the input frames, where traditional methods often fail to interpolate content accurately. Compared to other diffusion-based methods like LDMVFI and SVDKFI, Framer demonstrates superior adaptability to challenging cases and offers better control.

**Quantitative Comparison.** As discussed in VIDIM (Jain et al., 2024), reconstruction metrics like PSNR, SSIM, and LPIPS fail to capture the quality of interpolated frames accurately, since they penalize other plausible interpolation results that are not pixel-aligned with the original video. While generation metrics such as FID offer some improvement, they still fall short as they do not account for temporal consistency and evaluate frames in isolation. Despite this, we present the quantitative metrics for various settings on both datasets, where our method achieves the best FVD score among all baselines as in Table 1. We also evaluate Framer with 5 random point trajectories from ground-truth videos, estimated using Co-Tracker. As can be seen, "Framer with Co-Tracker" achieves superior performance even in reconstruction metric. For a more comprehensive assessment of quality, we recommend reviewing the supplementary comparison videos.

**User Study.** Since quantitative metrics fall short in reflecting video quality, we further assessed our method's performance through a user study. In this study, participants reviewed video sets generated from the same input frame pair by both existing methods and our Framer. Participants assessed up to 100 randomly ordered video sets and selected the one they found most realistic. In total, 20 participants provided 1,000 ratings across these video sets. As illustrated in Fig. 5, the results demonstrate a strong preference among human raters for the outputs produced by our method.

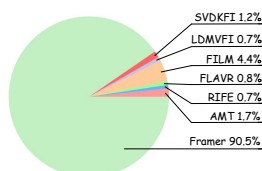

Figure 5: Reults on human preference.

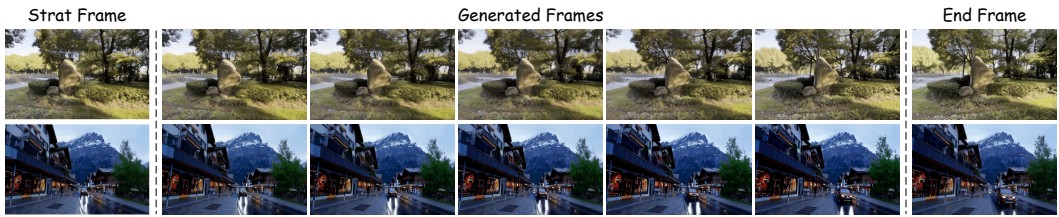

Figure 7: Novel view synthesis on both static (1st row) and dynamic scenes (2nd row).

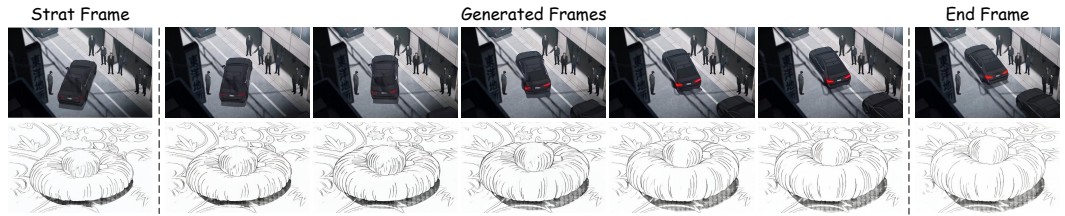

Figure 8: Applications on cartoon (1st row) and sketch (2nd row) interpolation.

## 4.3 APPLICATIONS

**Optional drag control.** Given the same input start and end frames, multiple plausible results can satisfy the goal of video interpolation. With Framer, users can direct the motion of the entities in input frames with simple drags for their intention, or simply obtain a default interpolation result without drags. As shown in Fig. 6, the seal moves in varying directions given the same input frames.

**Novel view synthesis (NVS)** is a classical 3D vision task, with a wide range of applications. Using images of different viewpoints as the start and end frames of the video respectively, we can realize the NVS from sparse viewpoint input by performing video interpolation. As shown in Fig. 7, our method achieves pleasing NVS in both static scenes (first row) and dynamic scenes (second and third rows). Taking the second row as an example, the house gradually moves out of the scene as the camera keeps moving forward. In the meantime, the car moves in the opposite direction to the camera and gradually takes up a larger proportion in the frame.

**Cartoon and sketch interpolation.** We can dramatically simplify the process of cartoon video production, by interpolating manually created cartoon images. To this end, we tested our method on cartoon data. Although our method is not specifically trained on cartoon videos, it produces appealing cartoon video results and supports both color images and sktech drawing frame interpolation, as shown in Fig. 8. For example, our method successfully models the motion of two objects, *i.e.*, the front vehicle pulls sideways while the rear vehicle follows, as shown in the first row. In the third row, Framer produces a smooth motion of the hand lifting in sketch drawings.

**Time-lapsing video generation.** Time-lapse photography can vividly demonstrate slow changes that are difficult to detect with the naked eye. Typically, it requires sufficient storage space to hold a large amount of image data. Video interpolation provides a simple and effective way to obtain time-lapse videos by interpolating frames with only a few images of key moments. As shown in Fig. 9, Framer produces the smooth change of moon waxing and waning.

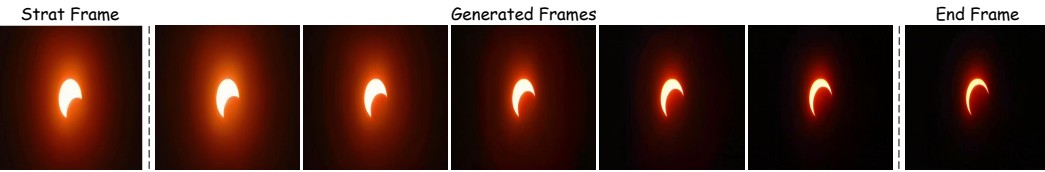

Figure 9: Applications on time-lapsing video generation.

Strat Frame Generated Frames End Frame y-t

Figure 10: Applications on slow-motion video generation. The y-t slice highlighted in red on video frames is visualized on the right.

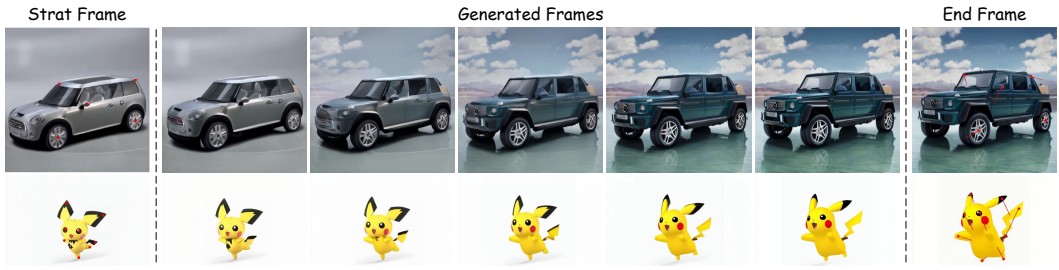

Strat Frame Generated Frames End Frame

Figure 11: Applications on image morphing. Customized trajectories ares overlaid on end frames.

**Slow-motion video generation** enhances visual effects by highlighting fine details and allows closer examination of fast phenomena. Our `Framer` inherently supports fast frame interpolation, as demonstrated in Fig. 10, enabling smooth slow-motion effects suitable for films and animations.

**Image morphing** (Aloraibi, 2023) is a popular image transformation technique with many applications in computer vision and computer graphics. Given two topologically similar images, it aims to generate a series of reasonable intermediate images. Using tue two images as the start and end frames, `Framer` can produce natural and smooth image morphing results. For example, in Fig. 1, we show the "evolution" process of Pokemon. More cases can be found in Fig. S16.

### 4.4 ABLATIONS STUDIES

We conducted ablation studies on the individual components of `Framer` to validate their effectiveness. The results are illustrated in Fig. 12. Our observations are as follows. First, when the trajectory guidance is removed (denoted as "w/o traj."), the foreground motorcycle exhibits significant distortion, as shown in the 1st row of Fig. 12. Conversely, with the inclusion of trajectory guidance, the temporal consistency of the video is notably enhanced, as depicted in the 2nd row. We believe this is due to the enhancement of point correspondence modeling across frames. Second, removing trajectory updates (denoted as "w/o traj. update") or updating the trajectory without bi-directional consistency checks (denoted as "w/o bi-directional") results in blurring in the wheel regions of the output video. We suspect the blurring is caused by the guidance of unnatural motion from inaccurate trajectories, which conflicts with the generation prior in the pre-trained diffusion model, leading to local blurring. In contrast, our method produces video frame interpolation results with natural motion and smooth temporal coherence. The quantitative results in Tables S1 and S2 in Appendix B further support these findings, showing a similar trend to the qualitative ablation experiments.

### 4.5 FAILURE CASES

Though `Framer` achieves superior performance of video frame interpolation, it still faces several limitations. Here we examine potential scenarios where the model may underperform or encounter failures, particularly in cases where it fails to capture object semantics, or no suitable correspondences between the input frames can be found. The failure cases are presented in Fig. 13.

**Failure to capture object semantics.** When a moving object appears blurred in the first and last frames of a sequence, it's sometimes difficult for the model to accurately interpret the object's semantics, which can lead to unrealistic generation results. As highlighted in DynamiCrafter (Xing et al., 2023), incorporating text guidance during video frame interpolation enhances the model's

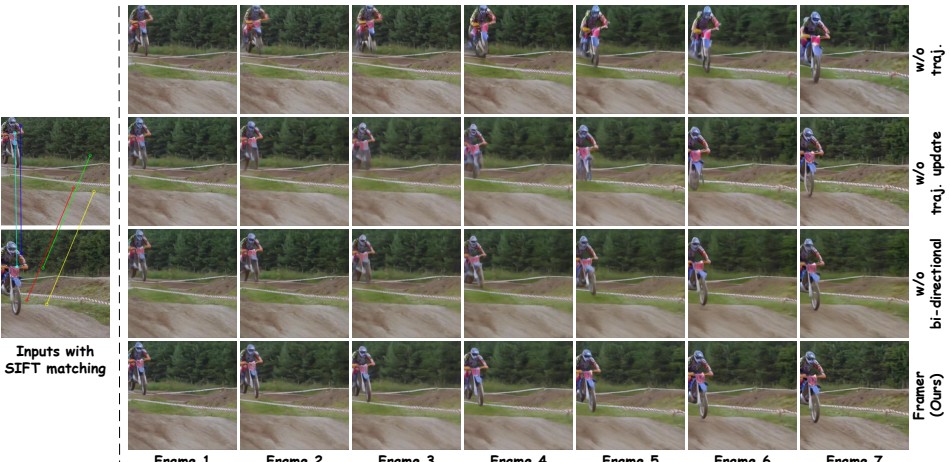

Figure 12: Ablations on each component. "w/o trajectory" denotes inference without guidance from point trajectory, "w/o traj. update" indicates inference without trajectory updates, and "w/o bi" suggests trajectory updating without bi-directional consistency verification.

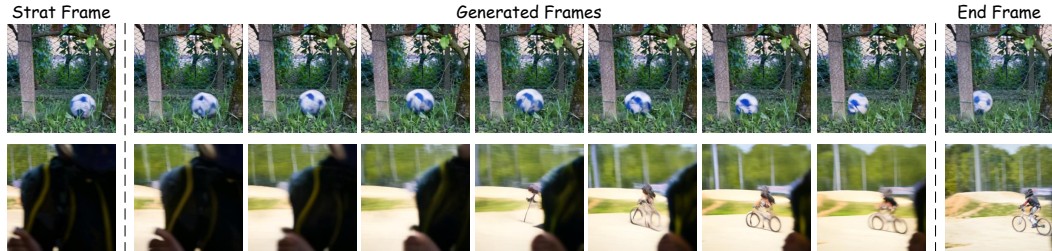

Figure 13: Examples of failure cases.

comprehension of moving objects and helps to resolve ambiguities associated with unclear objects. For this reason, we plan to introduce textual guidance to mitigate this problem in future work.

**Lack of suitable matching points between the first and last frames**. When there are no suitable matching points between the first and last frames, Framer struggles to utilize trajectory guidance effectively, resulting in sub-optimal video interpolation results. The model faces challenges when generating a scene where a character present in the initial frame exits while another character emerges from the background. As shown in Fig. 13, while Framer can produce reasonable video interpolation results, there is still noticeable distortion in the image. To address this problem, we are exploring the use of text guidance, as well as leveraging more advanced video models like Mochi (Team, 2024) and CogVideo-X (Yang et al., 2024), to improve our handling of such scenes.

## 5 CONCLUSION AND FUTURE WORK

In this paper, we introduce Framer, an interactive frame interpolation pipeline designed to produce smoothly transitioning frames between two images, guided by user-defined point trajectories. By harnessing user input point controls from the start and end frames, we effectively guide the video interpolation process. Moreover, our method offers an "autopilot" mode that introduces a module to automatically estimate keypoints and refine trajectories without manual input. Through extensive experiments and user studies, we demonstrate the superiority of our method in achieving promising results in terms of both the quality and controllability of the interpolated frames. However, challenges remain, particularly in transitioning between different clips. A potential solution involves splitting the clips into several keyframes and then interpolating these keyframes sequentially. Future work will focus on addressing these challenges.

## ACKNOWLEDGMENTS

This work is partially supported by the National Key R&D Program of China (NO.2022ZD0160160101), Ningbo Science and Technology Bureau (Grant Number 2024Z291) and the National Natural Science Foundation of China (No. 62206244). CS is the corresponding author.

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

APPENDIX

## A  MORE IMPLEMENTATION DETAILS

During training, we sample 14 consecutive frames from videos, with a spatial resolution of $512 \times 320$. Specifically, we center-crop the video to an aspect ratio of $512/320$, then resize the video frames to the resolution of $512 \times 320$. Random horizontal flip is utilized for data augmentation. We sample the video in temporal dimension, with a frame interval of 2. For the training of the point trajectory-based ControlNet, we sample 1 to 10 trajectories with larger motions for training. Specifically, we follow ReVideo (Mou et al., 2024a) and sample the trajectories by setting the normalized lengths of the trajectories as sampling probabilities. During "autopilot" mode sampling, we use the Euler sampler with 30 diffusion steps in total. For point tracking in Sec. 3.3, we use the output feature of the second decoder block in the 3D-UNet. We provide an Algorithm in Alg. 1 to illustrate the point trajectory estimation method. We resize the shorter side of the video to the length of 512, then center crop the video to the resolution of $512 \times 320$.

We transform 2D points into gaussian heatmaps, following the practice in Stacked Hourglass (Newell et al., 2016), DragNUWA (Yin et al., 2023), and DragAnthing (Wu et al., 2024). Specifically, we initialize a canvas map with the same height and width of the input video, setting all values to zero. Subsequently, for each trajectory point at the coordinate position $p$, we create a grid region centered on this point with a pixel area of 41x41. The center of this area (coordinate $p$) is assigned a value of 1, while the values decrease in accordance with a Gaussian distribution as the distance from $p$ increases. The variance of this Gaussian distribution is set to 8 in both the horizontal and vertical directions.

---

**Algorithm 1** Point trajectory estimation algorithm in "Autopilot mode".

---

**Input:** $I_0$: start image, $I_n$: end image
$m$: number of matching points
distance threshold for nearest neighbor search $r_1$
distance threshold for the bi-directional consistency check $r_2$

**Output:** $\{\boldsymbol{p}_i\}_{i=1}^m$: Anchor point list for $m$ trajectories
$c_{traj}$: Esitimated point trajectory

$\triangleright$ Apply SIFT point matching
$\{\boldsymbol{p}_i\}_{i=1}^m = \{[p_i^0, p_i^n]\}_{i=1}^m \leftarrow \mathrm{SIFT}(I_0, I_n)$     $\triangleright$ $\boldsymbol{p}_i$ denotes the known anchor points on the trajectory

$\triangleright$ Point trajectory updating
**for** $i = 1...m$ **do**
   $\triangleright$ Interpolate the anchor points to obtain the current trajectory
   $c_{traj} = \mathrm{Interpolate}(\boldsymbol{p}_i)$
   **for** $k = 1...n - 1$ **do**
      $\triangleright$ Forward point tracking
      $p_i^{k,0} := \underset{q_i^k \in \Omega(p_i^k, r_1)}{\arg\min} \left\| \mathbf{F}\left(q_i^k\right) - \mathbf{F}\left(p_i^0\right) \right\|_1$
      $\triangleright$ Backward point tracking
      $p_i^{k,n} := \underset{q_i^k \in \Omega(p_i^k, r_1)}{\arg\min} \left\| \mathbf{F}\left(q_i^k\right) - \mathbf{F}\left(p_i^n\right) \right\|_1$
      $\triangleright$ Bi-directional consistency check and point update
      **if** $p_i^{k,n} \in \Omega\left(p_i^{k,0}, r_2\right)$
      **then** $\boldsymbol{p}_i \leftarrow p_i^k = (p_i^{k,0} + p_i^{k,n})/2$     $\triangleright$ Add updated point to the anchor point list
   **end for**
**end for**

---

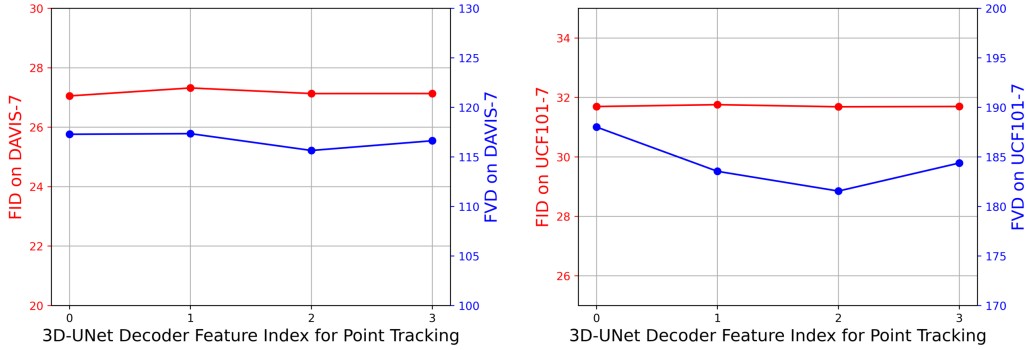

Figure S1: Ablations on diffusion feature for point tracking at test time, experiments conducted on DAVIS-7 (left) and UCF101-7 (right).

# B    MORE DETAILED ABLATION RESULTS

## B.1    QUALITATIVE RESULTS FOR ABLATION STUDIES.

In Fig. 12, we show the qualitative results for ablation studies. We supplement these results with the quantitative results in Table S1 and Table S2, which show a similar trend to the qualitative ablation experiments.

|  | DAVIS-7 | | | | | UCF101-7 | | | | |
|---|---|---|---|---|---|---|---|---|---|---|
|  | PSNR↑ | SSIM↑ | LPIPS↓ | FID↓ | FVD↓ | PSNR↑ | SSIM↑ | LPIPS↓ | FID↓ | FVD↓ |
| w/o trajectory | 20.19 | 0.6831 | 0.2787 | 28.25 | 128.71 | 24.16 | 0.8677 | 0.1798 | 32.64 | 195.54 |
| w/o traj. updating | 20.82 | 0.7054 | 0.2621 | 27.33 | 120.73 | 24.69 | 0.8748 | 0.1842 | 31.95 | 187.37 |
| w/o bi-directional | 20.94 | 0.7102 | 0.2602 | 27.23 | 116.81 | 24.73 | 0.8746 | 0.1845 | **31.66** | 183.74 |
| Framer (**Ours**) | **21.23** | **0.7218** | **0.2525** | **27.13** | **115.65** | **25.04** | **0.8806** | **0.1714** | 31.69 | **181.55** |

Table S1: Ablations on each component, evaluating all 7 generated frames. "w/o trajectory" denotes inference without guidance from point trajectory, "w/o traj. updating" indicates inference without trajectory updating, and "w/o bi" suggests trajectory updating without bi-directional consistency verification.

|  | DAVIS-7 (mid-frame) | | | | UCF101-7 (mid-frame) | | | |
|---|---|---|---|---|---|---|---|---|
|  | PSNR↑ | SSIM↑ | LPIPS↓ | FID↓ | PSNR↑ | SSIM↑ | LPIPS↓ | FID↓ |
| w/o trajectory | 19.30 | 0.6504 | 0.3093 | 57.10 | 23.14 | 0.8523 | 0.1967 | 54.98 |
| w/o traj. updating | 19.84 | 0.6700 | 0.2935 | 55.37 | 23.60 | 0.8590 | 0.2009 | 53.83 |
| w/o bi-directional | 19.95 | 0.6739 | 0.2919 | 54.75 | 23.65 | 0.8586 | 0.2016 | 53.54 |
| Framer (**Ours**) | **20.18** | **0.6850** | **0.2845** | **55.13** | **23.92** | **0.8646** | **0.1889** | **53.3**3 |

Table S2: Ablations on each component, evaluating only the middle frame out of all 7 generated frames. "w/o trajectory" denotes inference without guidance from point trajectory, "w/o traj. updating" indicates inference without trajectory updating, and "w/o bi" suggests trajectory updating without bi-directional consistency verification.

## B.2    ABLATIONS ON DIFFUSION FEATURE FOR POINT TRACKING.

As detailed in Sec. 3.3, we perform point tracking using the diffusion feature for point trajectory updating. Here we perform ablated experiments on the selection of the diffusion feature. The results are shown in Fig. S1. It can be seen that in both DAVIS-7 and UCF-7, point tracking with the output feature from the second diffusion block gives rise to the best-performing results in FVD.

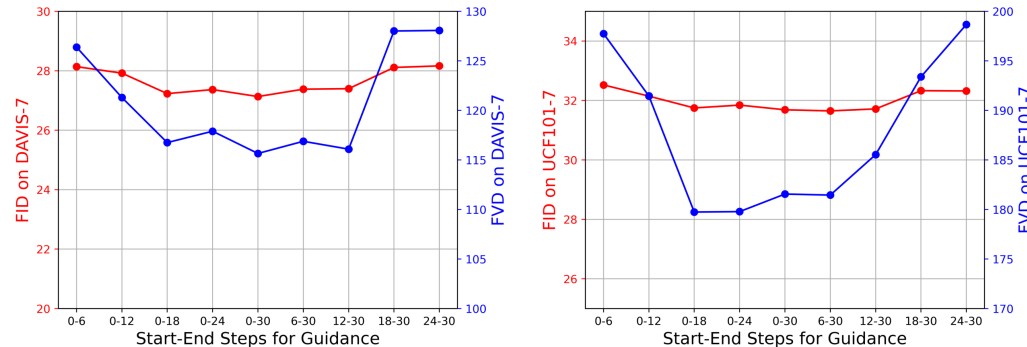

Figure S2: Ablations on the start and end diffusion steps for correspondence guidance, experiments conducted on DAVIS-7 (left) and UCF101-7 (right). We use a total sampling step of 30.

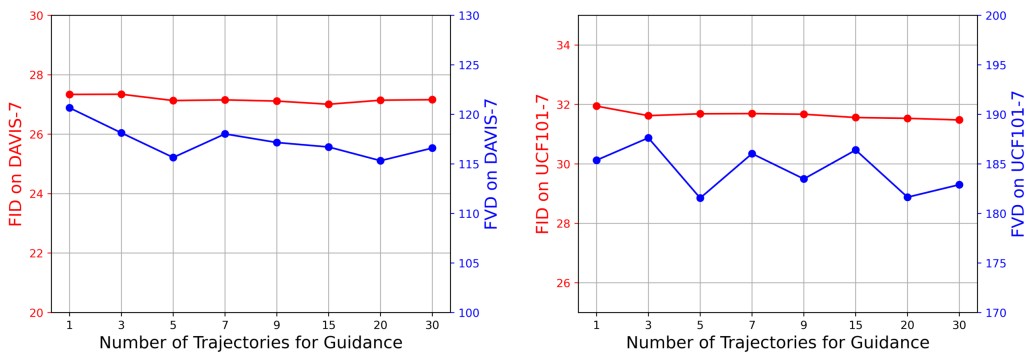

Figure S3: Ablations on the number of trajectories for guidance during sampling, experiments conducted on DAVIS-7 (left) and UCF101-7 (right).

### B.3 ABLATIONS ON DIFFUSION STEPS FOR CORRESPONDENCE GUIDANCE.

We ablate the diffusion steps for correspondence guidance by only applying the guidance at the early steps or late steps in diffusion sampling. The results are shown in Fig. S2. As can be seen, the early steps are often more important than the late steps for correspondence modeling. For example, on DAVIS-7, a pleasing FVD can be obtained when performing guidance only on 0-18 diffusion steps. By contrast, performing guidance only on 18-30 diffusion steps brings litter improvements. We speculate that this is because the early diffusion steps focus on the structural information of the video, while the late diffusion steps focus on the texture and details (Xue et al., 2023). The correspondence guidance at early steps already helps the model obtain a reasonable video structure. In the implementation, we simply apply correspondence guidance in all diffusion steps, without detailed searches on the hyper-parameter.

### B.4 ABLATIONS ON THE NUMBER OF TRAJECTORIES FOR CORRESPONDENCE GUIDANCE.

As described in Sec. 3.3, we use $m$ trajectories for correspondence guidance during sampling. Here we perform ablated experiments on this hyper-parameter, and the result is shown in Fig. S3. It can be seen that sampling with the 5 trajectories leads to the best performance. Thus we set $m = 5$ by default.

| | DAVIS-7 (mid-frame) | | | | UCF101-7 (mid-frame) | | | |
|---|---|---|---|---|---|---|---|---|
| | PSNR↑ | SSIM↑ | LPIPS↓ | FID↓ | PSNR↑ | SSIM↑ | LPIPS↓ | FID↓ |
| AMT (Li et al., 2023) | 20.59 | 0.6834 | 0.3564 | 100.36 | 25.24 | 0.8837 | 0.2237 | 75.97 |
| RIFE (Huang et al., 2020) | **20.74** | 0.6813 | 0.3102 | 80.78 | **25.68** | **0.8842** | 0.1835 | 59.33 |
| FLAVR (Kalluri et al., 2023) | 19.93 | 0.6514 | 0.4074 | 118.45 | 24.93 | 0.8796 | 0.2164 | 79.86 |
| FILM (Reda et al., 2022) | 20.28 | 0.6671 | **0.2620** | **48.70** | 25.31 | 0.8818 | **0.1623** | **41.23** |
| LDMVFI (Danier et al., 2024) | 19.87 | 0.6435 | 0.2985 | 56.46 | 25.16 | 0.8789 | 0.1695 | 43.01 |
| DynamicCrafter (Xing et al., 2023) | 14.61 | 0.4280 | 0.5082 | 77.65 | 17.05 | 0.6935 | 0.3502 | 97.01 |
| SVDKFI (Wang et al., 2024a) | 16.06 | 0.4974 | 0.3719 | 53.49 | 20.03 | 0.7775 | 0.2326 | 69.26 |
| Framer (**Ours**) | 20.18 | **0.6850** | 0.2845 | 55.13 | 23.92 | 0.8646 | 0.1889 | 53.33 |
| Framer with Co-Tracker (Ours) | 21.94 | 0.7693 | 0.2437 | 55.77 | 25.86 | 0.8868 | 0.1873 | 54.64 |

Table S3: Quantitative comparison with existing video interpolation methods on reconstruction and generative metrics, evaluated only on the middle frame out of all 7 generated frames.

| | PSNR↑ | SSIM↑ | LPIPS↓ | FID↓ | FVD↓ | NIQE↓ |
|---|---|---|---|---|---|---|
| 2x | 23.60 | 0.8203 | 0.1992 | 24.12 | N/A | 5.0753 |
| 4x | 23.08 | 0.7899 | 0.2091 | 25.92 | 93.42 | 4.9948 |
| 8x | 21.23 | 0.7218 | 0.2525 | 27.13 | 115.65 | 5.0598 |

Table S4: Evaluation on generating varying numbers of middle frames.

## C  MORE QUANTITATIVE RESULTS

### C.1  MORE DETAILS ON COMPARISON WITH PREVIOUS METHODS

We follow the practice of VIDIM (Jain et al., 2024) and perform the quantitative evaluation on DAVIS-7 and UCF101-7 datasets using both reconstruction and generative metrics. Both DAVIS-7 and UCF101-7 are obtained by sampling 7 consecutive video frames from the corresponding datasets. We use all videos in the DAVIS dataset and a subset of 400 videos in the UCF101 dataset.

In Table S3 we provide the quantitative comparison based on the middle frame of the 7 interpolated video frames. Besides, in Fig. S7, Fig. S8, Fig. S9, and Fig. S10, we show more qualitatively comparisons with exiting methods.

### C.2  GENERALIZATION TO VARYING NUMBER OF MIDDLE FRAMES

Framer supports generating a variable number of frames between the start and end frames. Video interpolation at different frame rates can be achieved by adjusting the number of temporal channels in the initial noise during sampling. Although the model is fine-tuned using 14 consecutive frames sampled from the training videos, it generalizes effectively to frame interpolation with varying numbers of intermediate frames. In the main text, we evaluated frame interpolation with 7 intermediate frames ($8\times$ in the temporal dimension) to align with the evaluation settings in VIDIM (Jain et al., 2024). Here, we extend our analysis to explore different settings for video frame interpolation, including $2\times$ and $4\times$. The results of these experiments are presented in Table S4. As shown in the table, Framer delivers pleasing results for video frame interpolation across different numbers of intermediate frames, highlighting its robustness.

### C.3  QUANTITATIVE RESULTS ON MORE BENCHMARKS

Following the practice of Zhong et al. (2024), we conduct experiments on the Vimeo90K septuplet dataset (Xue et al., 2019), X4K1000FPS (Sim et al., 2021), and Adobe240 (Su et al., 2017) to evaluate the performance of Framer. The results are shown in Table S5 and Table S6. It can be seen that Framer achieves competitive results on these datasets, especially on the NIQE metric, since it does not require video results to be pixel-aligned with the ground truth.

### C.4  RESULTS ON CARTOON INTERPOLATION

We collected 500 cartoon videos from the Internet to make comparisons with existing cartoon interpolation methods, including AnimeInterp (Siyao et al., 2021), EISAI (Chen & Zwicker, 2022), and ToonCraft (Xing et al., 2024). Following the practice of ToonCraft (Xing et al., 2024), we

| | PSNR↑ | SSIM↑ | LPIPS↓ | NIQE↓ |
|---|---|---|---|---|
| RIFE (Huang et al., 2020) | 28.22 | 0.912 | 0.105 | 6.663 |
| IFRNet (Kong et al., 2022) | 28.26 | 0.915 | 0.088 | 6.422 |
| AMT (Li et al., 2023) | 28.52 | 0.920 | 0.920 | 6.866 |
| EMA-VFI (Zhang et al., 2023a) | **29.41** | 0.928 | 0.086 | 6.736 |
| InterpAny-Clearer [D] (Zhong et al., 2024) | 29.20 | **0.929** | 0.092 | 6.475 |
| InterpAny-Clearer [D, R] (Zhong et al., 2024) | 28.84 | 0.926 | 0.081 | 6.286 |
| LDMVFI (Danier et al., 2024) | 27.43 | 0.912 | 0.092 | 6.279 |
| DynamiCrafter (Xing et al., 2023) | 26.51 | 0.891 | 0.128 | 6.912 |
| SVDKFI (Wang et al., 2024a) | 28.01 | 0.903 | 0.082 | 5.969 |
| Framer (**Ours**) | 28.32 | 0.918 | **0.072** | **5.623** |

Table S5: Quantitative results on Vimeo90K (Xue et al., 2019) septuplet dataset.

| | X4K1000FPS7 | | | | Adobe240 | | | |
|---|---|---|---|---|---|---|---|---|
| | PSNR↑ | SSIM↑ | LPIPS↓ | NIQE↓ | PSNR↑ | SSIM↑ | LPIPS↓ | NIQE↓ |
| RIFE (Huang et al., 2020) | 36.36 | **0.967** | 0.040 | 7.130 | 30.24 | **0.939** | 0.073 | 5.206 |
| InterpAny-Clearer [D] (Zhong et al., 2024) | **36.80** | 0.964 | 0.032 | 6.936 | **30.47** | 0.938 | 0.057 | 4.974 |
| InterpAny-Clearer [D, R] (Zhong et al., 2024) | 36.26 | 0.964 | **0.032** | 6.924 | 30.30 | 0.937 | **0.054** | **4.907** |
| LDMVFI (Danier et al., 2024) | 36.03 | 0.954 | 0.035 | 6.314 | 29.95 | 0.911 | 0.072 | 5.328 |
| DynamiCrafter (Xing et al., 2023) | 35.42 | 0.925 | 0.051 | 7.116 | 27.54 | 0.883 | 0.084 | 5.824 |
| SVDKFI (Wang et al., 2024a) | 36.31 | 0.938 | 0.046 | 6.621 | 28.43 | 0.903 | 0.069 | 5.695 |
| Framer (**Ours**) | 36.38 | 0.955 | 0.033 | **5.632** | 29.89 | 0.914 | 0.068 | 5.045 |

Table S6: Quantitative results on X4K1000FPS (Sim et al., 2021) and Adobe240 (Su et al., 2017) dataset.

tested on 512×320 resolution and each video contains 16 frames. The results are as follows. As can be seen, even though Framer is not specifically trained on cartoon data, it achieves superior results on this task. The qualitative comparison can be seen in Fig. S4. We found that when the difference between the start and end frames is large, existing methods often fail to connect the first and last frames, producing blurred results (Siyao et al., 2021; Chen & Zwicker, 2022), or directly jumping from the start frame to the end frame (Xing et al., 2024). In contrast, Framer can utilize the correspondence between input frames and the guidance of the trajectory to connect input frames, thus producing coherent videos.

## C.5 RESULTS ON SKETCH INTERPOLATION

Comparisons on the sketch interpolation task. We extracted the sketch of these cartoons using Anime2Sketch (Xiaoyu et al., 2021) to make comparisons with different methods. It can be seen that Framer achieves competitive performance, demonstrating its superiority in sketch interpolation. The qualitative comparison can be seen in Fig. S5.

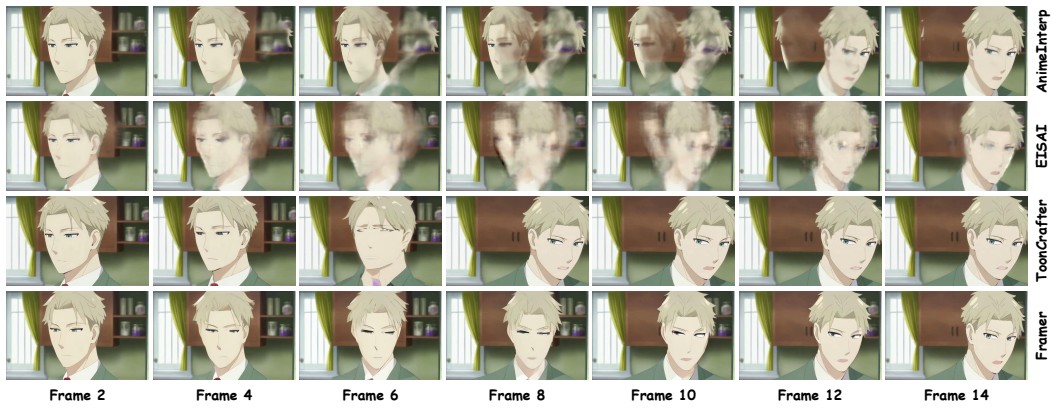

Figure S4: Qualitative comparison on cartoon interpolation.

| | PSNR↑ | SSIM↑ | LPIPS↓ | FID↓ | FVD↓ | NIQE↓ |
|---|---|---|---|---|---|---|
| AnimeInterp (Siyao et al., 2021) | 14.51 | 0.6196 | 0.4521 | 55.13 | 322.14 | 6.91 |
| EISAI (Chen & Zwicker, 2022) | 13.86 | 0.5142 | 0.4132 | 62.41 | 483.09 | 7.24 |
| ToonCrafter (Xing et al., 2024) | 16.34 | 0.5988 | 0.3576 | 34.74 | 208.34 | 6.68 |
| Framer (Ours) | **19.33** | **0.6591** | **0.2280** | **24.72** | **142.85** | **6.17** |

Table S7: Quantitative results on cartoon interpolation.

| | PSNR↑ | SSIM↑ | LPIPS↓ | FID↓ | FVD↓ | NIQE↓ |
|---|---|---|---|---|---|---|
| AnimeInterp (Siyao et al., 2021) | 16.77 | 0.4374 | 0.3263 | 55.13 | 281.54 | 12.13 |
| EISAI (Chen & Zwicker, 2022) | 15.25 | 0.3872 | 0.3513 | 73.08 | 352.72 | 12.75 |
| ToonCrafter (Xing et al., 2024) | 17.07 | 0.4767 | 0.2978 | 48.89 | 135.53 | 12.00 |
| AnimeInbet (Siyao et al., 2023) | **21.42** | 0.5829 | 0.2413 | 31.82 | 136.07 | **11.51** |
| Framer (Ours) | 20.88 | **0.6489** | **0.1237** | **28.63** | **106.48** | 12.53 |

Table S8: Quantitative results on sketch interpolation.

# D MORE QUALITATIVE RESULTS

## D.1 MORE QUALITATIVE RESULTS ON APPLICATIONS.

We provide more qualitative results on drag control, novel view synthesis, cartoon and sketch interpolation, time-lapsing video generation, slow-motion video generation, and image morphing in Fig. S11, Fig. S12, Fig. S13, Fig. S14, Fig. S15, and Fig. S16, respectively.

## D.2 QUALITATIVE RESULTS ON INTERPOLATING COMPLEX MOTIONS.

We additionally provide qualitative results in interpolation frames in complex scenarios with large motions, as shown in Fig. S6.

# E DISCUSSIONS ON LIMITATIONS

Framer is built on top of the large-scale pre-trained video diffusion model, thus it inherits the limitations of the pre-trained model. Moreover, the point trajectories in Framer rely on the matching points between the input image pair for interpolating complex motions. While this is a step forward compared with current models that can only simply motions, our method still faces difficulties when the differences between the front and back frames are so large that no matched points can be found at all. Thus, we will explore more powerful pre-trained video diffusion models, as well as training video interpolation models on larger-scale video data in the future. Lastly, our approach currently only supports drag control and does not explore other interaction methods. In the future, we will continue to explore other user-friendly controls such as text control and camera pose control.

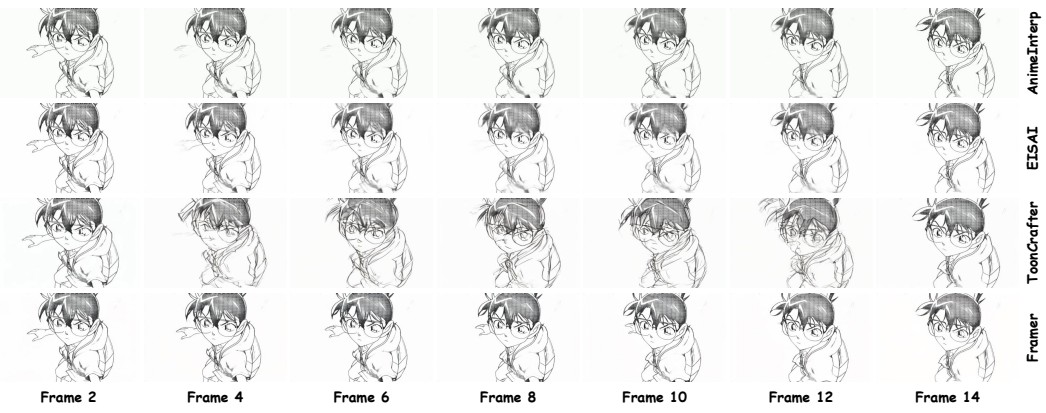

Figure S5: Qualitative comparison on sketch interpolation.

Strat Frame          Generated Frames          End Frame

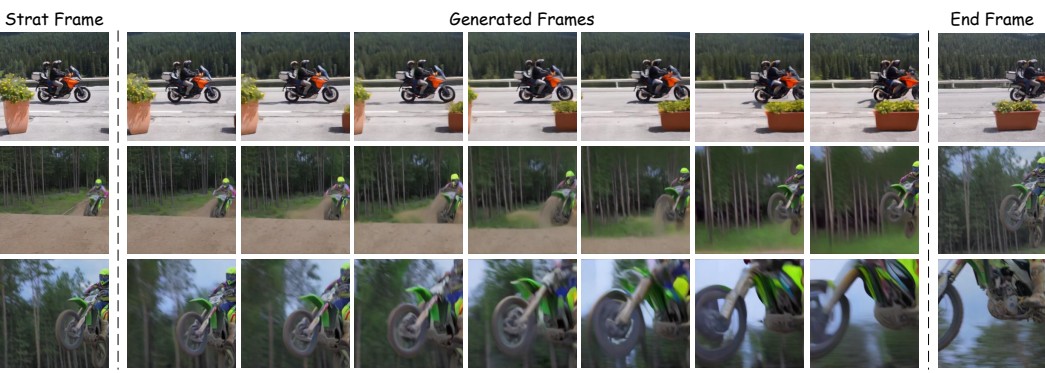

Figure S6: Results on input frames with large differences.

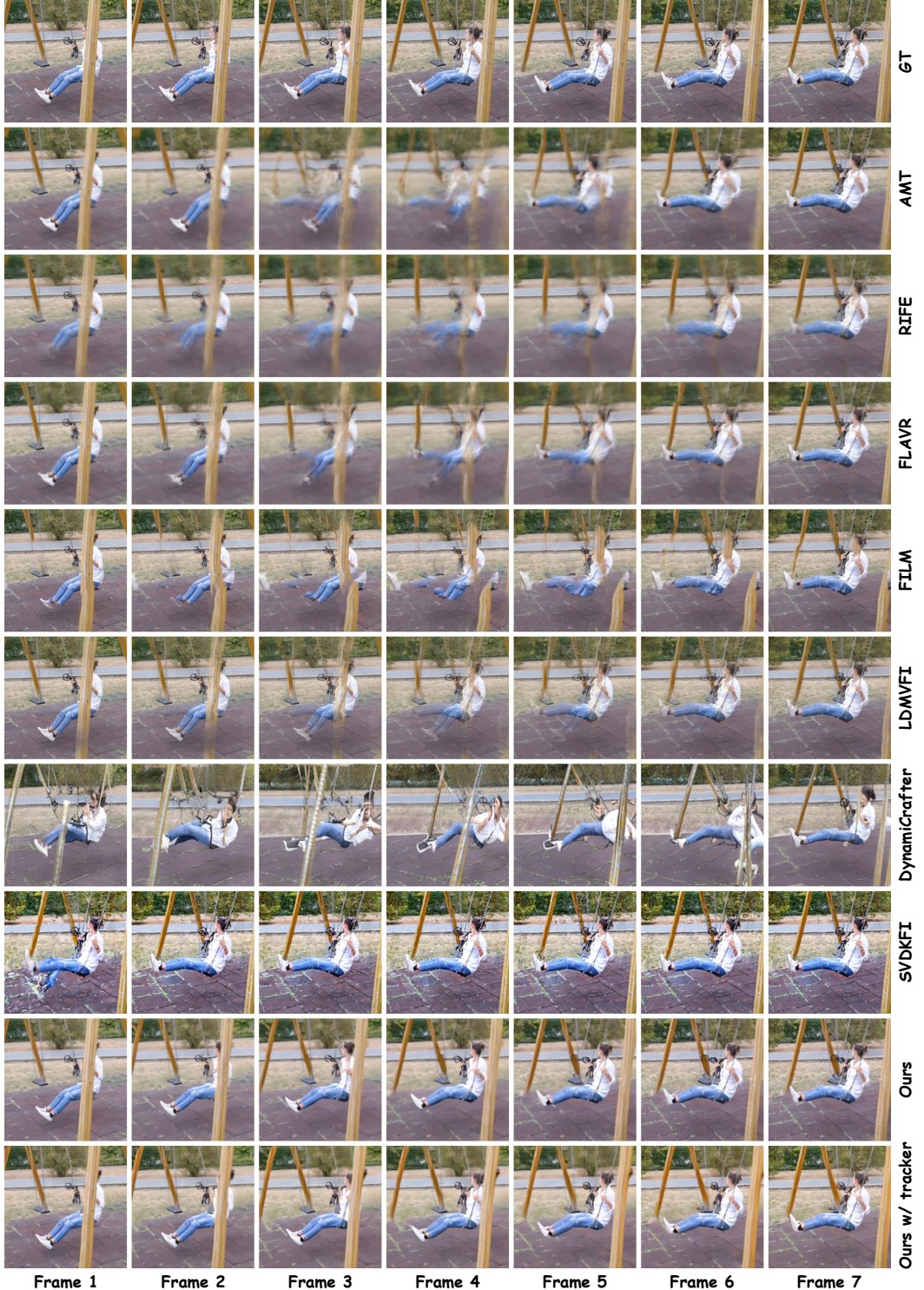

Figure S7: More qualitative comparison with existing methods. "GT" strands for ground truth.

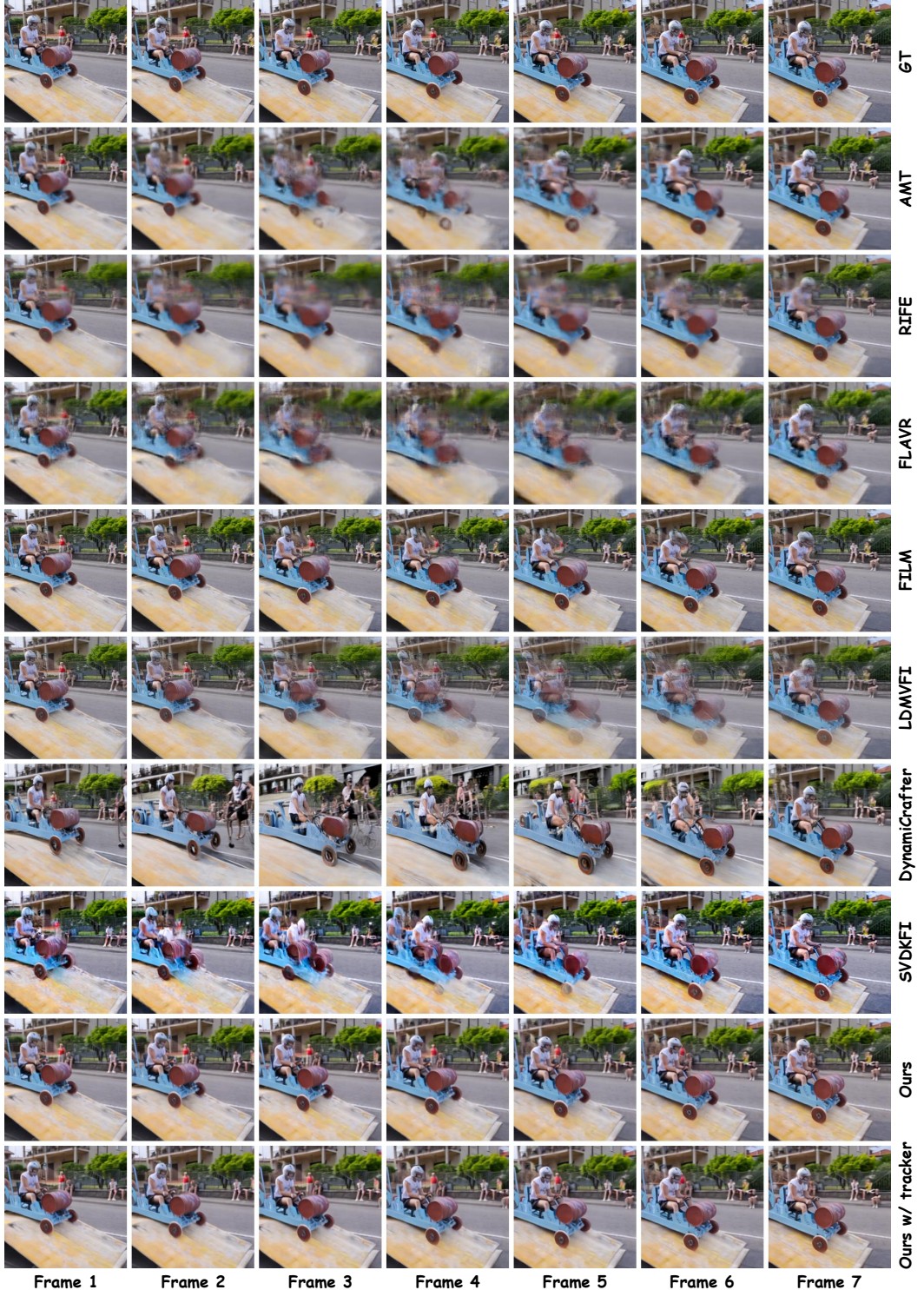

Figure S8: More qualitative comparison with existing methods. "GT" strands for ground truth.

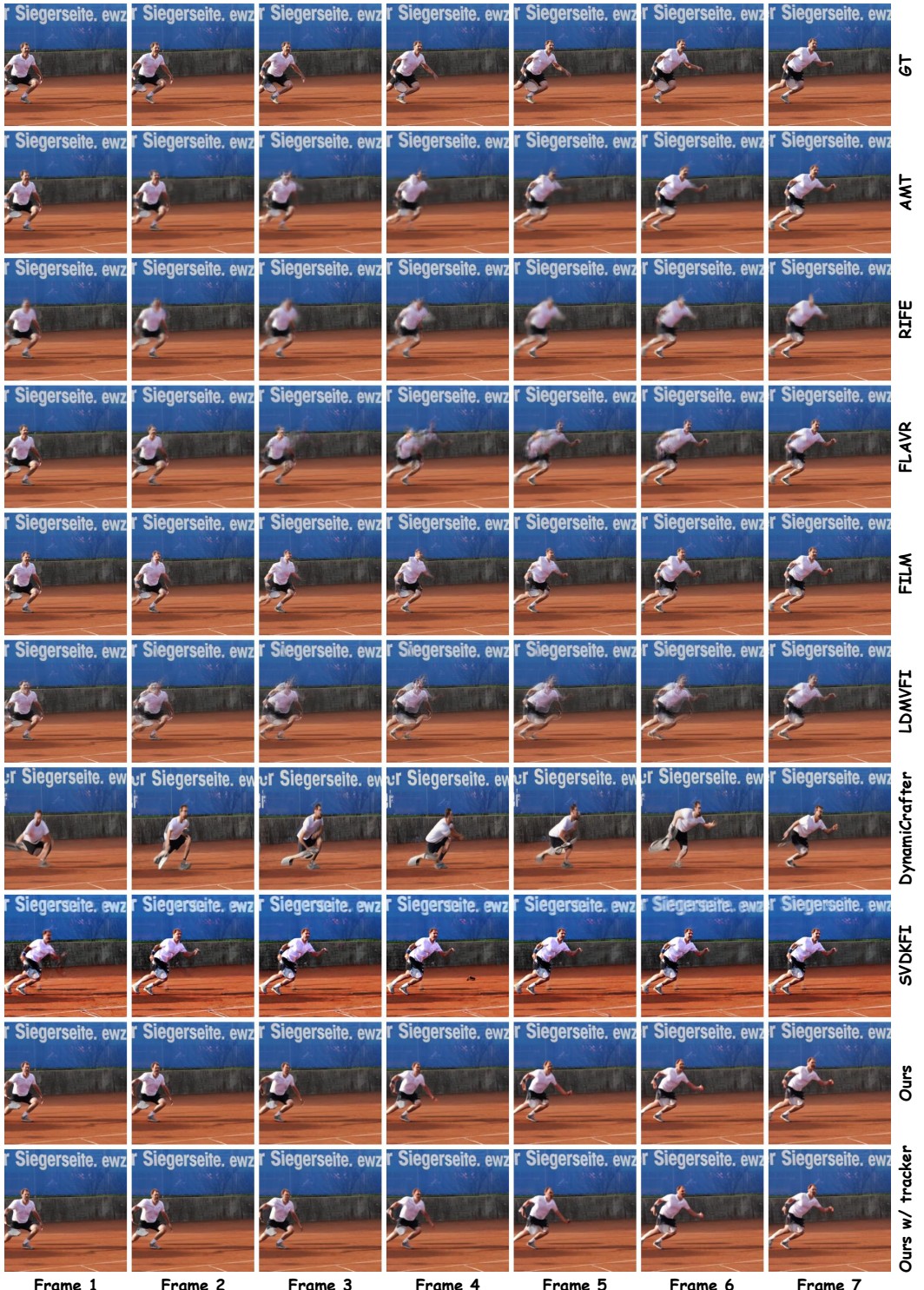

Figure S9: More qualitative comparison with existing methods. "GT" strands for ground truth.

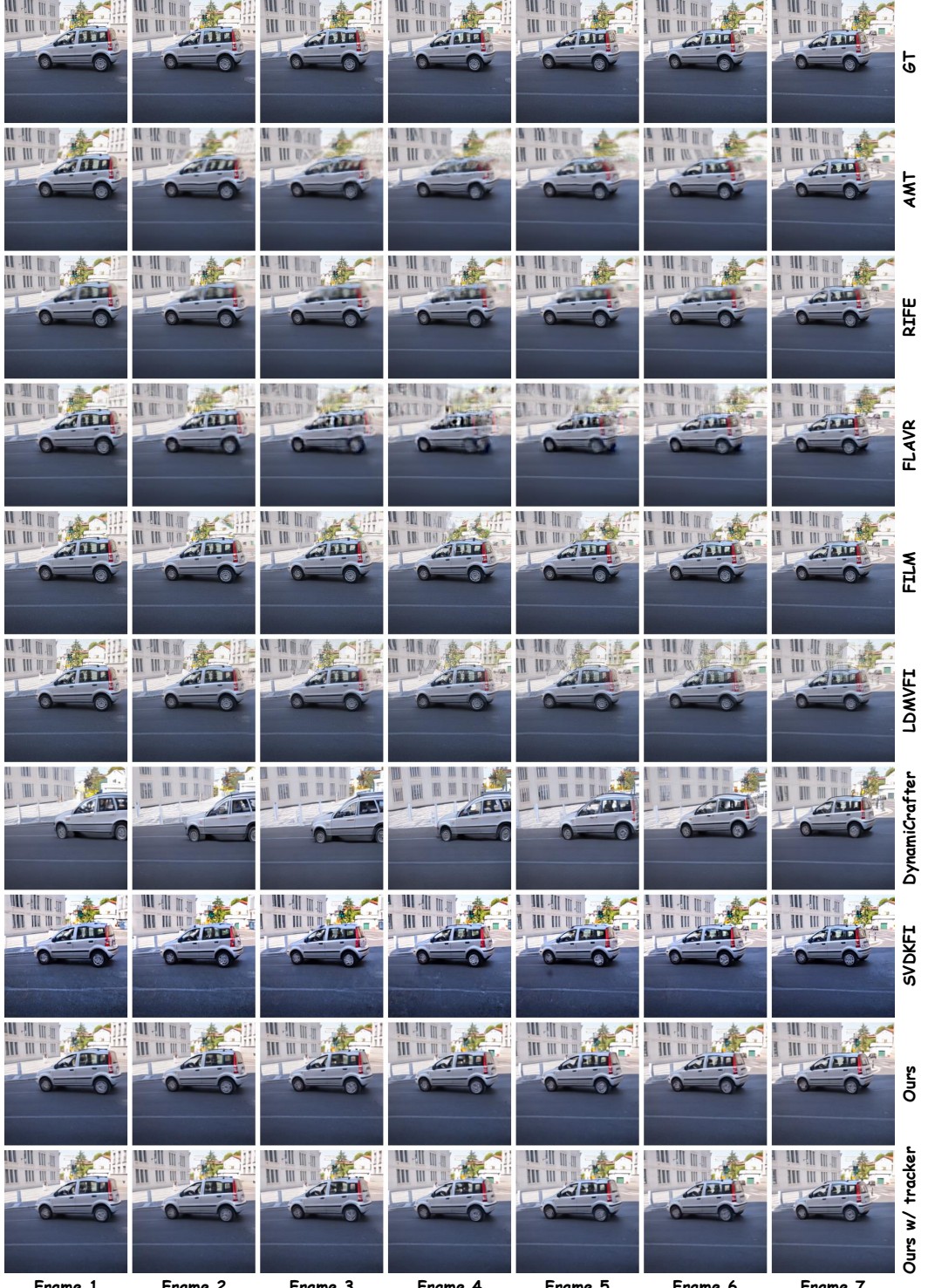

Figure S10: More qualitative comparison with existing methods. "GT" strands for ground truth.

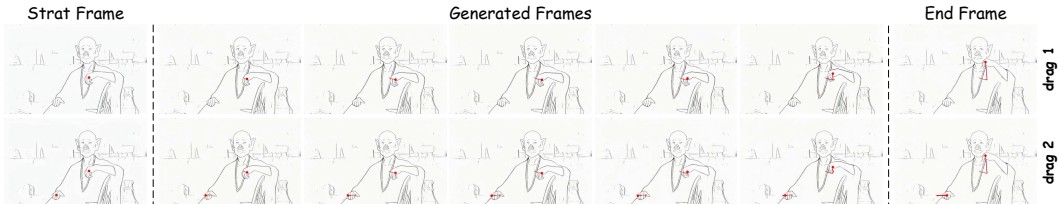

Figure S11: More results on user interaction. We show the results of two trajectory controls with the same input image pair.

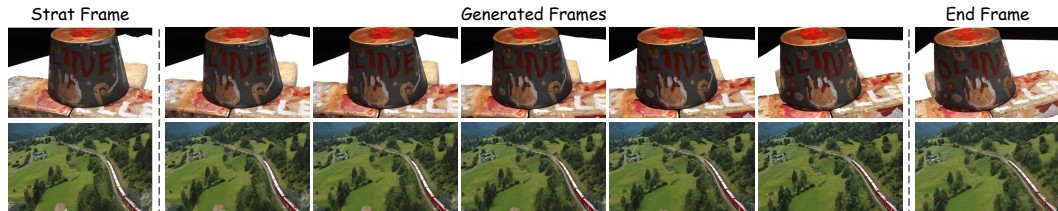

Figure S12: More results on novel view synthesis. The first and second rows show results on static and dynamic scenes, respectively.

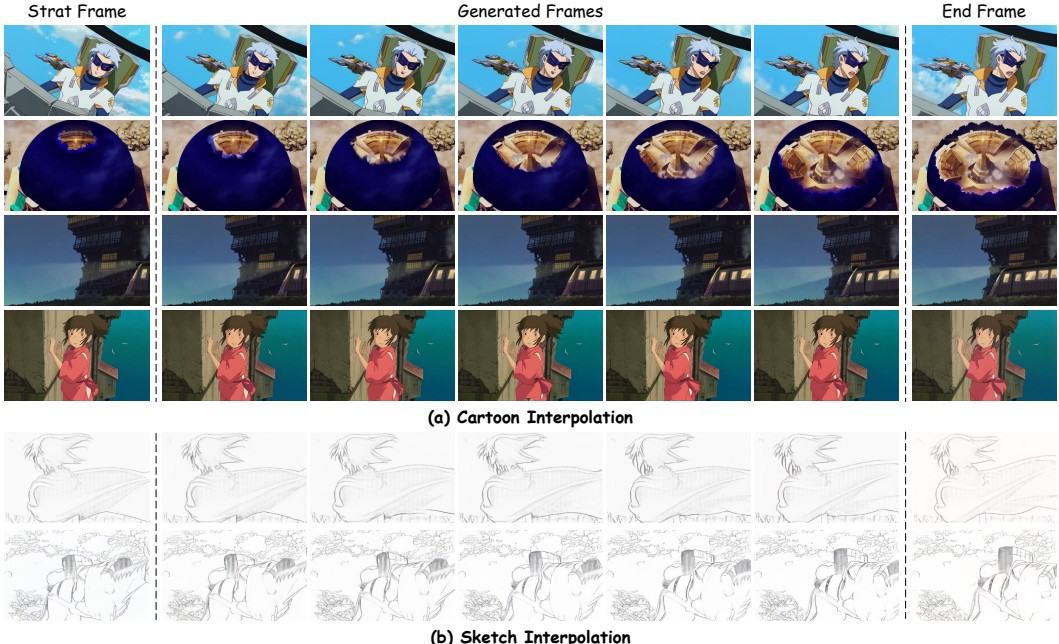

Figure S13: More results on (a) cartoon and (b) sketch interpolation.

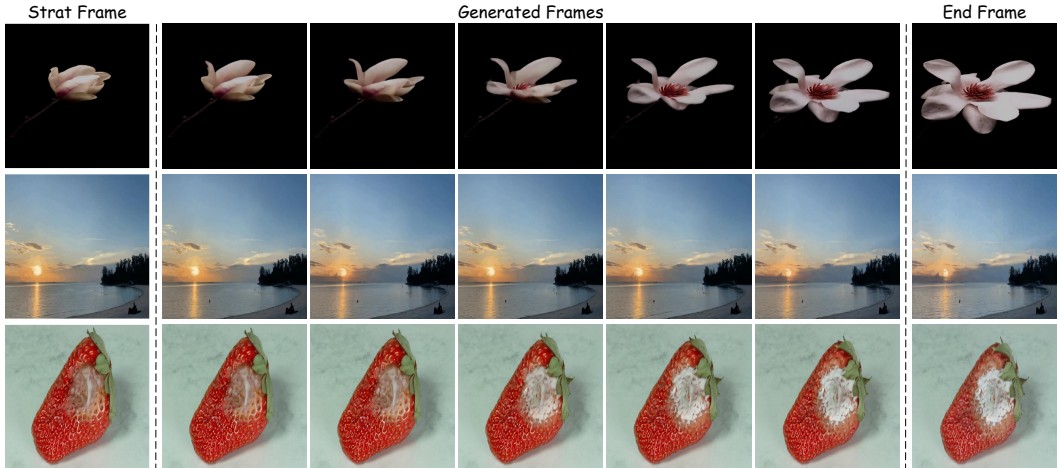

Figure S14: More results on time-lapsing video generation.

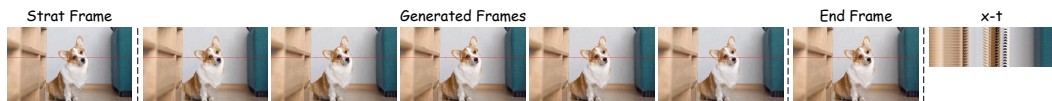

Figure S15: More results on slow-motion video generation. The x-t slice highlighted in red on video frames is visualized on the right.

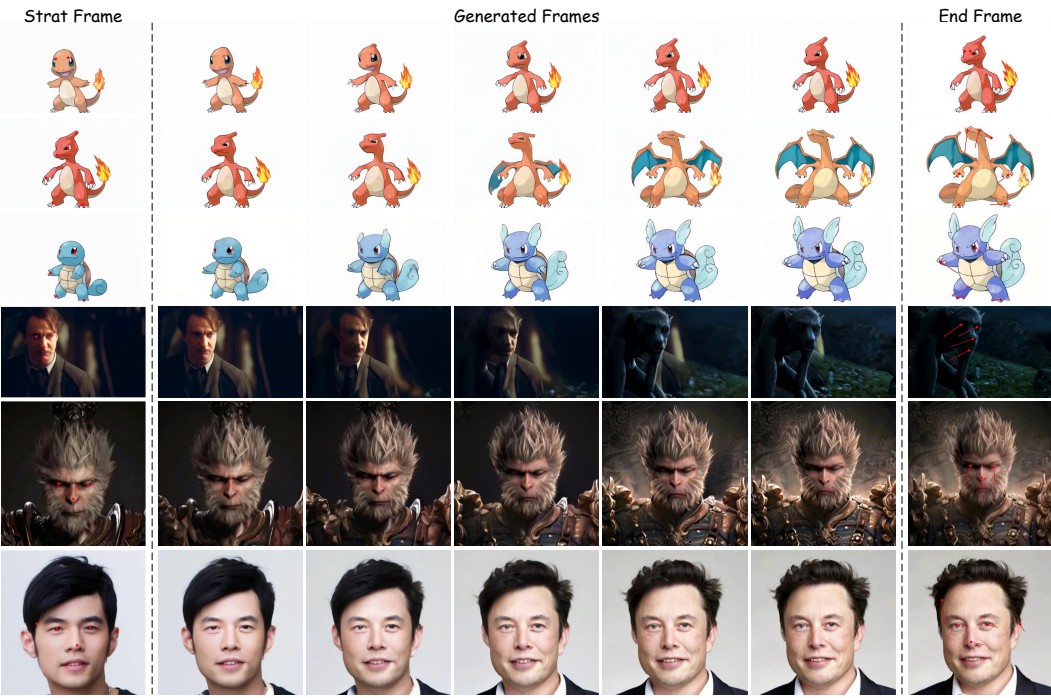

Figure S16: More results on image morphing.

