# OpenReview forum: "Framer: Interactive Frame Interpolation"
_ICLR.cc/2025/Conference — ICLR 2025 Poster_

### Official Review · Reviewer_hifJ · 2024-10-18

**Soundness:** 4
**Presentation:** 4
**Contribution:** 4
**Rating:** 8
**Confidence:** 4

**Summary:**

The paper proposes Framer, an interactive frame interpolation model designed to generate smooth frame transitions between two images. The approach allows users to customize the interpolation process by adjusting the motion trajectories of keypoints. This interactive feature provides precise control over local motions, improving the model’s ability to handle challenging scenarios. The model also includes an autopilot mode that automatically estimates keypoints and refines trajectories, simplifying the interpolation process for users.

**Strengths:**

1. The motivation of the paper is clear and well-founded. The authors address the inherent variability in intermediate frame generation and propose an effective solution by integrating keypoint control, which allows for a more tailored interpolation process that better aligns with user preferences.
2. The paper is well-written and easy to follow. The use of figures and tables is excellent, with ample illustrative examples demonstrating the model’s application.
3. The results presented are noticeably better than existing methods, particularly in challenging scenarios. Additionally, the inclusion of an automated mode ensures fairness in comparison, which further strengthens the validity of the results.

**Weaknesses:**

1. It would be helpful to see examples of failure cases, if any, to better understand the model’s limitations. Displaying such cases could provide insight into potential improvements or areas where the method might struggle in real-world applications.
2. The title might be slightly misleading. The approach seems more aligned with image morphing rather than traditional frame interpolation. A title that reflects this focus on image-to-image transitions could better represent the paper’s core contribution.

**Questions:**

Overall, I think this is a solid piece of work. Aside from the minor points mentioned in the weaknesses section, I don’t have significant concerns. The approach is interesting and well-presented, with promising results. I would lean toward recommending acceptance of this paper.

---

> ### Author Response · Authors · 2024-11-22
> **Responses to Reviewer hifJ**
>
> Thanks for your careful review and constructive suggestions. The questions are answered as follows.
>
> ---
>
> ### Q#1: It would be helpful to see examples of failure cases, if any, to better understand the model’s limitations.
>
> Thank you for pointing this out. Here we examine potential scenarios where the model may underperform or encounter failures, particularly in cases where it fails to capture object semantics, or no suitable correspondences between the input frames can be found. The failure cases are presented in Fig. 13 in the updated manuscript and the accompanying videos are presented in the Supplementary Material.
>
> **(a) Fail to capture object semantics.**
> When a moving object appears blurred in the first and last frames of a sequence, it's sometimes difficult for the model to accurately interpret the object's semantics, which can lead to unrealistic generation results. As highlighted in DynamiCrafter [1], incorporating text guidance during video frame interpolation enhances the model's comprehension of moving objects and helps to resolve ambiguities associated with unclear objects. For this reason, we plan to introduce textual guidance to mitigate this problem in future work.
>
> **(b) Lack of suitable matching points between the first and last frames.**
> When there are no suitable matching points between the first and last frames, Framer struggles to utilize trajectory guidance effectively, resulting in sub-optimal video interpolation results. The model faces challenges when generating a scene where a character present in the initial frame exits while another character emerges from the background. As shown in Fig. 13, while Framer can produce reasonable video interpolation results, there is still noticeable distortion in the image. To address this problem, we are exploring the use of text guidance, as well as leveraging more advanced video models like Mochi [3] and CogVideo-X [4], to improve our handling of such scenes.
>
> We have added new sections to the paper to discuss the above issues, as shown in Sec. 4.5. In the future, we will use a stronger base model for video generation and introduce the guidance of text to achieve higher quality video frame interpolation.
>
> [1] DynamiCrafter: Animating Open-domain Images with Video Diffusion Priors. ECCV 2024.
> [2] ToonCrafter: Generative Cartoon Interpolation. SIGGRAPH Asia 2024.
> [3] Mochi 1: A new SOTA in open-source video generation models.
> [4] CogVideoX: Text-to-Video Diffusion Models with An Expert Transformer, Arxiv 2024.
>
>
> ---
>
> ### Q#2: The title might be slightly misleading. The approach seems more aligned with image morphing rather than traditional frame interpolation.
>
> Thanks to your valuable suggestions, this paper aims at generative video frame interpolation and proposes an interactive frame interpolation approach based on point trajectories. We utilize automatically generated trajectories to enhance the performance of automated frame interpolation.
> The results show that Framer achieves superior performance in image morphing. However, we consider this task as a downstream application of generative video frame interpolation, among other downstream tasks like cartoon and sketch interpolation.
> For this reason, we would like to change the title to "Interactive Generative Frame Interpolation" to emphasize the generative nature of our method and to highlight downstream applications like image morphing.
> We welcome further discussion and suggestions.

---

> ### Author Response · Authors · 2024-11-25
> **Welcome Further Feedbacks**
>
> Dear Reviewer **hifJ**,
> We would like to thank you for the thoughtful and constructive feedback. During the rebuttal, we performed additional experiments and analyses to address your concerns. The experiments and analyses in our response are summarized as follows.
> - We discuss potential failure cases for Framer, including:
>   - Lack of suitable matching points between the first and last frames.
>   - Failure to capture object semantics.
> - We discuss a better title to emphasize the generative nature of Framer and to highlight downstream applications like image morphing.
>
>
> We are wondering if the responses have addressed your concerns and would appreciate it if you considered raising your final rating. Looking forward to your further feedback!

---

### Official Review · Reviewer_4487 · 2024-10-25

**Soundness:** 3
**Presentation:** 4
**Contribution:** 2
**Rating:** 6
**Confidence:** 5

**Summary:**

This paper presents a diffusion-based architecture, *Framer*, for interactive frame interpolation, introducing last-frame conditioning and keypoint trajectories to achieve controllable transitions. While the qualitative results are impressive, the practical application, related work discussion, and unique technical contributions need further improvement.

**Strengths:**

The method’s introduction of keypoint trajectories addresses motion ambiguity, leveraging the stable video diffusion model to improve interpolation fidelity.
The qualitative outcomes are notably strong.

**Weaknesses:**

1. **Related Work**: The paper does not sufficiently discuss previous work tackling motion ambiguity and interactive frame interpolation, such as [1] and [2], which address motion/velocity ambiguities. And [2] is the early work in interactive VFI. A more comprehensive discussion would clarify the novelty and contributions of *Framer*.
2. **Real-World Application**: The practical applicability of this method is unclear, especially given that real-world videos typically run at frame rates like 24–30fps, making extremely large-motion interpolation less common. There is also no analysis of inference speed in autopilot mode.
3. **Robustness and Scope**: The method’s effectiveness under complex trajectories remains uncertain, as the learned keypoint trajectory distribution is unlikely to generalize to all cases. Additionally, autopilot mode’s reliance on feature-matching algorithms may limit performance under conditions like blur or low resolution. Further experiments are needed to clarify performance under these situations.
4. **Benchmarking and Quantitative Analysis**: The paper lacks evaluations on commonly used VFI datasets, such as Vimeo90K and X4K1000FPS, which would offer valuable comparison. In addition, considering that the proposed method achieves superior performance only in terms of FVD, the inclusion of non-reference metrics such as NIQE will help to further understand the image quality.
5. **Technical Innovation**: While the method yields high-quality visuals, the technical novelty appears limited. Techniques such as conditioning diffusion models with start and end frames and using keypoint trajectories (DragGAN) already exist. Emphasizing unique contributions would strengthen the paper’s impact.

[1] "Exploring motion ambiguity and alignment for high-quality video frame interpolation." CVPR2023
[2] "Clearer Frames, Anytime: Resolving Velocity Ambiguity in Video Frame Interpolation." ECCV2024.

**Questions:**

Please address the issues raised in the weaknesses section. This reviewer is willing to increase the rating if the issues are well addressed.

---

> ### Author Response · Authors · 2024-11-22
> **Responses to Reviewer 4487 (1/3)**
>
> Thanks for your careful review and constructive suggestions. The questions are answered as follows.
>
> ---
> ### Q#1 Related Work: The paper does not sufficiently discuss previous work tackling motion ambiguity and interactive frame interpolation.
>
> Thanks for pointing this out. Realizing that motion ambiguity remains given the start and end frames, [1] proposes a texture consistency loss to encourage predictions that maintain similar structures with their counterparts in the given frames. On the other hand, [2] resolves the ambiguities by providing explicit hints on how far objects have traveled between start and end frames, termed "distance indexing".
> While progress has been made, we emphisize the differences with our work from several perspectives. **Firstly**, the motivation of [1] and [2] is to resolve motion ambiguity in video frame interpolation, while our approach prioritizes interaction and managing significant disparities for generative frame interpolation. The difference in motivation leads to different frameworks for VFI. While [1] and [2] are flow-based, we built upon generative video diffusion models. **Secondly**, while [2] also supports user-interaction, the distance-indexing interaction requires setting detailed distance values for middle frames, making them less intuitive. By contrast, our interaction is simple and intuitive. Users can realize the interaction by simply dragging points on images. **Thirdly**, [2] leverages object-level representation for distance indexing, by setting a constant distance value for an entire object. This design makes it less effective when handling non-rigid object, since difference parts in object run in varying directions. By contrast, we use point trajectory control, allowing customing motions for different parts of an object. **Lastly**, for the application, [1] and [2] focus on improving the temporal rate of input videos, while we additionally emphasize user creativity and control over the interpolation process, enabling applications such as image morphing and cartoon interpolation beyond traditional interpolation techniques.
> The discussion has been integrated to the revised manuscript.
>
>
> [1] Exploring Motion Ambiguity and Alignment for High-Quality Video Frame Interpolation. CVPR 2023.
> [2] Clearer Frames, Anytime: Resolving Velocity Ambiguity in Video Frame Interpolation. ECCV 2024.
>
>
> ---
>
>
> ### Q#2: Real-World Application: The practical applicability of this method is unclear, especially given that real-world videos typically run at frame rates like 24–30fps, making extremely large-motion interpolation less common.
>
> Thank you for your valuable feedback. Existing methods still struggle to handle the fast and complex motions of non-rigid objects, such as the motion of a human running (see Fig. S6). Framer's interactive generative video frame interpolation **enables several applications beyond traditional video frame interpolation**, including but not limited to: (a) **cartoon and sketch interpolation**, which can significantly reduce the labor required for anime creators; (b) **Image Morphing**, which produces interesting and visually appealing effects; \(c) **Novel view synthesis** from sparse viewpoint inputs. Additionally, connecting two keyframes through generative video interpolation opens up **potential applications in video generation** [1], which merits further exploration in future work.
>
>
> |   Method           |  Time per video  |  Time per frame  |
> |:------------------:|:--------:|:-------:|
> |   Framer           |  4.64s   |  0.66s  |
> | LDMVFI [2]         |  9.34s   |  1.33s  |
> | DynamiCrafter [3]  |  13.16s  |  1.88s  |
> |   SVDKFI [4]       |  42.92s  |  6.13s  |
>
>
> We evaluate the inference speed on the DAVIS-7 dataset, using a A6000 GPU. The results are presented in the above Table. The results indicate that Framer's inference speed is competitive compared to other diffusion model-based video frame interpolation methods. However, we acknowledge that Framer currently does not achieve real-time video frame interpolation. In future iterations, we plan to leverage the latest advances in diffusion model pruning [5] and sampling acceleration [6] techniques to enhance Framer's inference speed.
>
>
> [1] MovieDreamer: Hierarchical Generation for Coherent Long Visual Sequences, Arxiv 2023.
> [2] LDMVFI: Video Frame Interpolation with Latent Diffusion Models, AAAI 2024.
> [3] DynamiCrafter: Animating Open-domain Images with Video Diffusion Priors, ECCV 2024.
> [4] Generative Inbetweening: Adapting Image-to-Video Models for Keyframe Interpolation, Arxiv, 2024.
> [5] AsyncDiff: Parallelizing Diffusion Models by Asynchronous Denoising, NeurIPS 2024.
> [6] One-step Diffusion with Distribution Matching Distillation, CVPR 2024.

---

> ### Author Response · Authors · 2024-11-22
> **Responses to Reviewer 4487 (2/3)**
>
> ---
>
> ### Q#3: Robustness and Scope: The method’s effectiveness under complex trajectories remains uncertain. Additionally, autopilot mode’s reliance on feature-matching algorithms may limit performance.
>
>
> Thank you for your suggestion. To validate the generation quality of Framer across various trajectory controls, we introduce **ObjMC [1] as a metric for assessing the accuracy of generated results in relation to trajectory adherence**. ObjMC calculates the Euclidean distance between real and predicted trajectories in the generated video. We further propose the Normalized ObjMC metric which normalizes the Euclidean distance by the length of the ground-truth trajectory. In implementation, we utilize Co-Tracker to extract trajectories from randomly sampled points in the DAVIS-7 dataset, measuring the complexity of these trajectories by their lengths. The trajectories for guidance are categorized into three groups based on the average distance between trajectory points in neighboring frames: short (0-10 pixel distance), medium (10-20 pixel distance), and long (more than 20 pixel distance).
>
> |        |  PSNR  |  SSIM  |  LPIPS  |  FID  |  FVD  |  NIQE  |    ObjMC    | Normalized ObjMC |
> |:------:|:------:|:------:|:-------:|:-----:|:-----:|:------:|:-----------:|:-----------:|
> |  shot  |  22.41  |  0.7794  |  0.2279  |  27.42  |  105.20  |  5.019  |  **3.734**  |  **0.2506**  |
> | medium |  22.31  |  0.7754  |  0.2283  |  27.47  |  104.70  |  5.022  |  **4.491**  |  **0.2460**  |
> |  long  |  22.23  |  0.7714  |  0.2297  |  27.32  |  108.50  |  5.002  |  **5.740**  |  **0.2720**  |
>
> The results indicate that Framer performs well when guided by trajectories of varying lengths. Moreover, the performance is stable when guided by trajectories of varying complexity.
>
> When using autopilot mode to process low-quality input images, it’s important to clarify that there is a match confidence threshold in SIFT to filter out unreliable matches. **If the number of matched points is below the default number of trajectories $m=5$ or if no matches are found, we use fewer tracks or no track guidance to avoid misguidance of low-quality matches.** To assess Framer's performance under extreme conditions, **we conducted downsampling on the videos in the DAVIS-7 dataset to create low-resolution inputs.** We then evaluated Framer across datasets with varying resolutions, summarizing the results in the table below.
>
> | Resolution |  PSNR  |  SSIM  |  LPIPS  |  FID  |   FVD  |  NIQE  |
> |:-----:|:------:|:------:|:-------:|:-----:|:------:|:------:|
> |  256  |  21.23 | 0.7218 |  0.2525 | 27.13 | 115.65 |  5.0598  |
> |  128  |  22.74 | 0.7806 |  0.2254 | 25.11 | 108.52 |  6.3905  |
> |  64   |  24.67 | 0.8388 |  0.2119 | 25.69 | 132.64 |  7.2161  |
>
> The results indicate that Framer maintains satisfactory performance even with low-resolution inputs.
>
>
> ---
>
>
> ### Q#4: The paper lacks evaluations on commonly used VFI datasets, such as Vimeo90K and X4K1000FPS. The inclusion of non-reference metrics such as NIQE will help to further understand the image quality.
>
>
> Thanks for your valuable suggestions. We conduct experiments on the Vimeo90K septuplet dataset, X4K1000FPS, and Adobe240 to evaluate the performance of Framer. The results as shown in the following three tables respectively. It can be seen that Framer achieves competitive results on these datasets, especially on the NIQE metric, since it does not require video results to be pixel-aligned with the ground truth.
>
> **Results on Vimeo90K septuplet dataset.**
> |  Method  |  PSNR  |  SSIM  |  LPIPS  |  NIQE  |
> |:-----:|:------:|:------:|:-------:|:------:|
> |    RIFE   |  28.22  |  0.912  |  0.105  |  6.663  |
> |    IFRNet |  28.26  |  0.915  |  0.088  |  6.422  |
> |    AMT    |  28.52  |  0.920  |  0.920  |  6.866  |
> |  EMA-VFI  |  **29.41**  |  **0.928**  |  0.086  |  6.736  |
> |  Framer   |  28.32  |  0.918  |  **0.072**  | **5.623** |
>
>
> **Results on X4K1000FPS.**
> |  Method  |  PSNR  |  SSIM  |  LPIPS  |  NIQE  |
> |:-----:|:------:|:------:|:-------:|:------:|
> |  RIFE  |  36.36  |  **0.967**  |  0.040  |  7.130  |
> |  Framer  |  **36.38**  |  0.955  | **0.033**  |  **5.632**  |
>
>
> **Results on Adobe240.**
> |  Method  |  PSNR  |  SSIM  |  LPIPS  |  NIQE  |
> |:-----:|:------:|:------:|:-------:|:------:|
> |  RIFE  |  **30.24**  |  **0.939**  |  0.073  |  5.206  |
> |  Framer |  29.89  |  0.914  |  **0.068**  |  **5.045** |
>
> Additionally, we conducted comparisons of NIQE metrics on the DAVIS-7 and UCF101-7 datasets. As shown in the below table, Framer achieves competitive results on NIQE.
>
> |  Dataset  |  AMT | RIFE |   FLAVR  |  FILM  |  LDMVFI  |  DynamiCrafter  |  SVDKFI  |  Framer  |
> |:-----:|:------:|:------:|:-----:|:------:|:-----:|:------:|:------:|:-------:|
> |  DAVIS-7  |  5.5708 | 5.2166 |   5.6361  |  5.3113  |  5.1062  |  6.1512  |  5.3949  | **5.0598**  |
> |  UCF101-7  |  5.6804 | 5.6671 |   5.5919  |  **5.1769**  |  5.3103  |  5.1139  |  5.5244  |  5.2342  |

---

> ### Author Response · Authors · 2024-11-22
> **Responses to Reviewer 4487 (3/3)**
>
> ---
>
> ### Q#5: Technical Innovation: While the method yields high-quality visuals, the technical novelty appears limited.
>
> Thanks for pointing this out. We highlight two key technical contributions introduced in Framer. Firstly, we utilize **explicit long-term correspondences between start and end frames** to enhance video frame interpolation (VFI). The popular flow-based methods rely on short-term temporal correlations between adjacent frames, while the kernel-based and recent diffusion-based approaches implicitly learn these correlations from training data, neglecting the explicit relationships between input frames. Consequently, these methods struggle with input frames that exhibit significant differences. In contrast, our approach leverages long-term temporal guidance derived from user drag inputs or estimated point trajectories, resulting in improved quality and temporal consistency in the output videos.
> Secondly, we present **a novel point trajectory estimation method that integrates feature matching with a new bi-directional point tracking technique** to achieve accurate point trajectory guidance. **Although a uni-directional point tracking method has been proposed in DragGAN, directly applying it to video frame interpolation leads to sub-optimal results**, as demonstrated in Section 4.4 and Table S1. To tackle this, our motivation is that matched keypoints in both start and end frames offer distinctive features for effective point tracking. Based on this observation, we implement both forward and backward point tracking and introduce a bi-directional consistency check process to enhance tracking accuracy.
> Beyond these technical contributions, Framer introduces **a novel way of interaction in video frame interpolation to align with user intentions**. Users can unleash their creativity through simple and intuitive click-and-drag operations on input images. Based on these, Framer supports novel downstream applications, such as image morphing and cartoon or sketch interpolation, which require user customization and the ability to manage large differences between input frames.

---

> > ### Comment · Reviewer_4487 · 2024-11-24
> >
> > Thanks for the authors' effort in addressing the reviewers’ concerns. While some issues have been resolved, there are still areas that require further clarification and improvement:
> >
> > 1. Applications Beyond "Traditional" Video Frame Interpolation
> >
> > The authors claim that "Framer's interactive generative video frame interpolation enables several applications beyond traditional video frame interpolation", and considerable space is devoted to demonstrating these abilities. However, the qualitative and quantitative comparisons focus solely on "traditional" video frame interpolation methods. To substantiate the claims, the manuscript should include comparisons and discussions with related works in cartoon and sketch interpolation, image morphing, and novel view synthesis.
> >
> > 2. Dataset and Method Comparisons
> >
> > The revised manuscript highlighted that "Following the practice of Zhong et al.(2024), we conduct experiments on the Vimeo90K septuplet dataset (Xue et al., 2019), X4K1000FPS (Simet al., 2021), and Adobe240 (Su et al., 2017) to evaluate the performance of Framer." However, to provide a more comprehensive evaluation, comparisons should also include Zhong et al. (2024) itself. Additionally, similar to the tables comparing results on DAVIS-7 and UCF101-7, incorporating comparisons with diffusion-based methods such as LDMVFI and SVDKFI on these datastes (Vimeo90K septuplet, X4K1000FPS, Adobe240) are necessary for a clearer and more complete understanding.
> >
> > 3. Inference Speed Analysis
> >
> > The table summarizing inference speed should be added to the manuscript. This should include both diffusion-based methods and “traditional” video frame interpolation methods to give readers a clearer perspective on performance efficiency.
> >
> > These additions and refinements will strengthen the manuscript and provide a more thorough evaluation of Framer's capabilities.

---

> ### Author Response · Authors · 2024-11-25
> **Further Responses to Reviewer 4487 (1/2)**
>
> Thanks for recognizing our efforts and thank you again for your valuable comments. We continue to address your other concerns below.
>
> ---
>
> ### Applications Beyond "Traditional" Video Frame Interpolation
> We address your concerns in the following four aspects. **The new experiments results will be added to the revised manuscript.**
>
> - We add a paragraph to our related work, discussing the application of video frame interpolation as follows. "Video frame interpolation has a wide range of applications in many fields. While traditional interpolation methods focus on improving the frame rate of the input video [1, 2, 3, 4], generative frame interpolation methods are more concerned with dealing with situations where the input frames have large differences [5, 6]. In addition, some works train tailored video frame interpolation models for specific application scenarios, such as cartoon frame interpolation [7, 8, 9], sketch frame interpolation [10, 11], etc. In this paper, we show that Framer can handle all of the above tasks under a unified framework, and allow users to achieve fine-grained control of the interpolation process through simple interactions."
>
> - Comparisons on the cartoon interpolation task. We collected 500 cartoon videos from the Internet to make comparisons with existing cartoon interpolation methods, including AnimeInterp [7], EISAI [8], and ToonCraft [9]. Following the practice of ToonCraft [9], we tested on 512x320 resolution and each video contains 16 frames. The results are as follows. As can be seen, even though Framer is not specifically trained on cartoon data, it achieves superior results on this task.
> We found that when the difference between the start and end frames is large, existing methods often fail to connect the first and last frames, producing blurred results [7, 8], or directly jumping from the start frame to the end frame [9]. In contrast, Framer can utilize the correspondence between input frames and the guidance of the trajectory to connect input frames, thus producing coherent videos.
>     |      |  PSNR   |  SSIM    |  LPIPS   |    FID  |    FVD   |  NIQE   |
>     |:----:|:-------:|:--------:|:--------:|:-------:|:--------:|:-------:|
>     | AnimeInterp [7] |  14.51   |  0.6196    |  0.4521   |    55.13  |    322.14   |  6.91   |
>     |    EISAI [8]    |  13.86   |  0.5142    |  0.4132   |    62.41  |    483.09   |  7.24   |
>     |    ToonCrafter [9]|  16.34   |  0.5988    |  0.3576   |    34.74  |    208.34   |  6.68   |
>     |    Framer       |  **19.33**   | **0.6591**    |  **0.2280**   |    **24.72**  |    **142.85**   |  **6.17**  |
>
> - Comparisons on the sketch interpolation task. We extracted the sketch of these cartoons using Anime2Sketch [12] to make comparisons with different methods. It can be seen that Framer achieves competitive performance, demonstrating its superiority in sketch interpolation.
>
>     |      |  PSNR   |  SSIM    |  LPIPS   |    FID  |    FVD   |  NIQE   |
>     |:----:|:-------:|:--------:|:--------:|:-------:|:--------:|:-------:|
>     | AnimeInterp [7] |  16.77   |  0.4374    |  0.3263   |    55.13  |    281.54   |  12.13   |
>     |    EISAI [8]    |  15.25   |  0.3872    |  0.3513   |    73.08  |    352.72   |  12.75   |
>     |    ToonCrafter [9]|  17.07   |  0.4767    |  0.2978   |    48.89  |    135.53   |  12.00   |
>     |   AnimeInbet [10]|  **21.42**  |  0.5829    |  0.2413   |    31.82  |    136.07   |  **11.51**   |
>     |    Framer       |  20.88   |   **0.6489**    |  **0.1237**   |    **28.63**  |    **106.48**   |  12.53   |
>
> - For the novel view synthesis (NVS) task, we note that Framer currently does not support camera pose control, making it difficult to compare it with existing methods for NVS. In the future, we will further introduce camera pose control and make a systematic comparison with the NVS method.
>
>
> [1] AMT: All-Pairs Multi-Field Transforms for Efficient Frame Interpolation, CVPR 2023.
> Deep Animation Video Interpolation in the Wild, CVPR 2021.
> [2] RIFE: Real-Time Intermediate Flow Estimation for Video Frame Interpolation, ECCV 2022.
> [3] FLAVR: Flow-Agnostic Video Representations for Fast Frame Interpolation, WACV 2023.
> [4] FILM: Frame Interpolation for Large Motion, ECCV 2022.
> [5] DynamiCrafter: Animating Open-domain Images with Video Diffusion Priors, ECCV 2024.
> [6] Video Interpolation With Diffusion Models, CVPR 2024.
> [7] Deep Animation Video Interpolation in the Wild, CVPR 2021.
> [8] Improving the Perceptual Quality of 2D Animation Interpolation, ECCV 2022.
> [9] ToonCrafter: Generative Cartoon Interpolation, SIGGRAPH Asia 2024.
> [10] Deep Geometrized Cartoon Line Inbetweening, ICCV 2023.
> [11] Bridging the Gap: Sketch-Aware Interpolation Network for High-Quality Animation Sketch Inbetweening, ACM MM 2024.
> [12] Anime2Sketch: A Sketch Extractor for Anime Arts with Deep Networks, GitHub 2021.
>
> ---

---

> ### Author Response · Authors · 2024-11-25
> **Further Responses to Reviewer 4487 (2/2)**
>
> ---
>
>
> ### Dataset and Method Comparisons
> Thanks to your valuable suggestions, we have supplemented the comparison with InterpAny-Clearer [1], LDMVFI [2], DynamiCrafter [3] and SVDKFI [4] on Vimeo90K septuplet, X4K1000FPS, Adobe240 and the results are shown below. The InterpAny-Clearer is implemented based on RIFE, and the [D] and [R] notation denote the distance indexing paradigm and iterative reference-based estimation, respectively. **The results have been updated to Tab. S4 and Tab. S5.**
>
> **Results on Vimeo90K septuplet dataset.**
> |  Method  |  PSNR  |  SSIM  |  LPIPS  |  NIQE  |
> |:-----:|:------:|:------:|:-------:|:------:|
> |    RIFE   |  28.22  |  0.912  |  0.105  |  6.663  |
> |    IFRNet |  28.26  |  0.915  |  0.088  |  6.422  |
> |    AMT    |  28.52  |  0.920  |  0.920  |  6.866  |
> |  EMA-VFI  |  **29.41**  |  0.928  |  0.086  |  6.736  |
> |  InterpAny-Clearer [D]  |  29.20  |  **0.929**  |  0.092  |  6.475  |
> |  InterpAny-Clearer [D, R]  |  28.84  |  0.926  |  0.081  |  6.286  |
> |  LDMVFI             |  27.43  |  0.912  |  0.092  |  6.279  |
> |  DynamiCrafter      |  26.51  |  0.891  |  0.128  |  6.912  |
> |  SVDKFI             |  28.01  |  0.903  |  0.082  |  5.969  |
> |  Framer   |  28.32  |  0.918  |  **0.072**  | **5.623** |
>
>
> **Results on X4K1000FPS dataset.**
> |  Method  |  PSNR  |  SSIM  |  LPIPS  |  NIQE  |
> |:-----:|:------:|:------:|:-------:|:------:|
> |  RIFE  |  36.36  |  **0.967**  |  0.040  |  7.130  |
> |InterpAny-Clearer [D]|  **36.80**  |  0.964  |  0.032  |  6.936  |
> |InterpAny-Clearer [D, R]|  36.26  |  0.964  |  **0.032**  |  6.924  |
> |  LDMVFI             |  36.03  |  0.954  |  0.035  |  6.314  |
> |  DynamiCrafter      |  35.42  |  0.925  |  0.051  |  7.116  |
> |  SVDKFI             |  36.31  |  0.938  |  0.046  |  6.621  |
> |  Framer  |  36.38  |  0.955  | 0.033  |  **5.632**  |
>
>
> **Results on Adobe240 dataset.**
> |  Method  |  PSNR  |  SSIM  |  LPIPS  |  NIQE  |
> |:-----:|:------:|:------:|:-------:|:------:|
> |  RIFE  |  30.24  |  **0.939**  |  0.073  |  5.206  |
> |InterpAny-Clearer [D]|  **30.47**  |  0.938  |  0.057  |  4.974  |
> |InterpAny-Clearer [D, R]|  30.30  |  0.937  |  **0.054**  |  **4.907**  |
> |  LDMVFI             |  29.95  |  0.911  |  0.072  |  5.328  |
> |  DynamiCrafter      |  27.54  |  0.883  |  0.084  |  5.824  |
> |  SVDKFI             |  28.43  |  0.903  |  0.069  |  5.695  |
> |  Framer |  29.89  |  0.914  |  0.068  |  5.045 |
>
>
> It can be seen that Framer is able to achieve competitive results and is able to achieve the best quantitative results among the diffusion model based methods.
>
> [1] Clearer Frames, Anytime: Resolving Velocity Ambiguity in Video Frame Interpolation, ECCV 2024.
> [2] LDMVFI: Video Frame Interpolation with Latent Diffusion Models, AAAI 2024.
> [3] DynamiCrafter: Animating Open-domain Images with Video Diffusion Priors, ECCV 2024.
> [4] Generative Inbetweening: Adapting Image-to-Video Models for Keyframe Interpolation, Arxiv, 2024.
>
> ---
> ### Inference Speed Analysis
> Thanks to your valuable suggestion, we have added the comparison of inference speeds to **Tab. 1 of the updated paper**. The results are presented as follows.
>
>
> |  Method  |  AMT | RIFE |   FLAVR  |  FILM  |  LDMVFI  |  DynamiCrafter  |  SVDKFI  |  Framer  |
> |:-----:|:------:|:------:|:-------:|:------:|:-----:|:------:|:------:|:-------:|
> |  Time per video |  0.165s | 0.072s |   0.028s  |  0.291s  |  9.340s  |  13.166s |  42.923s |  4.644s  |
> |  Time per frame |  0.024s | 0.010s |   0.004s  |  0.042s  |  1.334s  |  1.881s  |  6.132s  |  0.663s  |
>
>
> It can be seen that Framer is able to achieve the best inference speed in the diffusion model-based approachs, but still has a gap from the traditional video frame interpolation model. In future iterations, we plan to leverage the latest advances in diffusion model pruning [1], parallel inference [2] and sampling acceleration [3] techniques to enhance Framer's inference speed.
>
> [1] Cross-Attention Makes Inference Cumbersome in Text-to-Image Diffusion Models, Arxiv 2024.
> [2] AsyncDiff: Parallelizing Diffusion Models by Asynchronous Denoising, NeurIPS 2024.
> [3] One-step Diffusion with Distribution Matching Distillation, CVPR 2024.
>
>
> ---

---

> ### Author Response · Authors · 2024-11-27
> **Looking Forward to Further Discussions**
>
> The quantitative comparisons for cartoon and sketch interpolation have been updated in Tables S7 and S8 in the Appendix. Additionally, qualitative comparisons are presented in Figures S4 and S5 in the Appendix, accompanied by corresponding videos in the Supplementary Materials.
> We are wondering if the responses have addressed your concerns and would appreciate it if you considered raising your final rating. Looking forward to your further feedback!

---

> > ### Comment · Reviewer_4487 · 2024-11-27
> >
> > Thanks for the additional experiments.
> > Although the performance and inference efficiency still need to be improved.
> > However, this reviewer believes that this paper demonstrates the potential of diffusion models for video frame interpolation.
> > Therefore, this reviewer has improved the score.

---

> ### Author Response · Authors · 2024-11-27
>
> Thank you for your feedback! We are pleased that our response has addressed your concerns. We also appreciate your thoughtful reviews, which have significantly enhanced the quality of our paper.

---

### Official Review · Reviewer_SBQc · 2024-11-02

**Soundness:** 3
**Presentation:** 3
**Contribution:** 2
**Rating:** 6
**Confidence:** 3

**Summary:**

This paper presents an interactive frame interpolation framework to produce smoothly transitioning frames between two images, interactively or automatically. By harnessing user input point controls from the start and end frames, this method can effectively generate the video frames for different applications including drag control, novel view synthesis, cartoon and sketch interpolation, time-lapsing video generation, slow-motion video generation, and image morphing.

**Strengths:**

1）	The method is straight forward but effective, which can be used for many applications;
2）	Most results shown in this paper are impressive.
3）	Ablation experiments are sufficient to evaluate the method.

**Weaknesses:**

1)	The component of the Framer includes Trajectory preprocessing, Trajectory control and Interpolation. Although the method is effective, the novelty is limited. Trajectory control is designed by following DragNUWA, and the Interpolation is mainly based on SVD that determines the performance of this method.
2)	The examples shown in this paper are a bit simple. The motion and the difference between two key frames (first and last frames) are not complex. More complex scenarios, as cases shown in the paper “Training Weakly Supervised Video Frame Interpolation with Events”, should be considered to demonstrate the superiority of this method.
3)	The image quality of some results shown in this paper degrades compared to the first input frame. For example, the IQ of frames in “laugh.mp4” is worse than that of the start and end frames;
4)	The reconstruction metric of this method is not best when comparing it to related methods.

**Questions:**

1)	Can the frame interpolation function support arbitrary number of frames generation between the start and end inputs?
2)	how to process the case when the number of corresponding points between frames is too few?

---

> ### Author Response · Authors · 2024-11-22
> **Responses to Reviewer SBQc (1/2)**
>
> Thanks for your careful review and constructive suggestions. The questions are answered as follows.
>
> ---
>
> ### Q#1: Can the frame interpolation function support arbitrary number of frames generation between the start and end inputs?
>
> Yes, Framer supports generating a variable number of frames between the start and end frames. Video interpolation at different frame rates can be achieved by adjusting the number of temporal channels in the initial noise during sampling. Although the model is fine-tuned using 14 consecutive frames sampled from the training videos, it generalizes effectively to frame interpolation with varying numbers of intermediate frames.
>
> In our initial submission, we evaluated frame interpolation with 7 intermediate frames (8x in the temporal dimension) to align with the evaluation settings in VIDIM [1]. Here, we extend our analysis to explore different settings for video frame interpolation, including 2x and 4x. The results of these experiments are presented below.
>
> |      |  PSNR   |  SSIM    |  LPIPS   |    FID  |    FVD   |  NIQE   |
> |:----:|:-------:|:--------:|:--------:|:-------:|:--------:|:-------:|
> |  2x  |  23.60  |  0.8203  |  0.1992  |  24.12  |  N/A     |  5.0753  |
> |  4x  |  23.08  |  0.7899  |  0.2091  |  25.92  |   93.42  |  4.9948  |
> |  8x  |  21.23  |  0.7218  |  0.2525  |  27.13  |  115.65  |  5.0598  |
>
>
> As shown in the table, our method delivers pleasing results for video frame interpolation across different numbers of intermediate frames, highlighting its robustness.
>
>
> [1]  Video Interpolation With Diffusion Models, CVPR 2024.
>
>
> ---
>
> ### Q#2: how to process the case when the number of corresponding points between frames is too few?
>
> Thank you for highlighting this point. We admit that Framer still faces difficulties when there is no correspondence between the first and last frames, although such situations are relatively rare in video interpolation scenarios. In these cases, the model cannot take advantage of the correspondence guidance from point trajectories, but relies more on the generative capabilities of the pre-trained video diffusion model to achieve effective frame interpolation.
> To enhance our approach, we plan to explore training Framer with more advanced video generation models, such as Mochi [1] and CogVideo-X [2], to leverage their generative strengths. Additionally, we will explore incorporating per-point visibility annotations on the trajectory condition $c_{traj}$ during both training and sampling. By labeling the visibility of points on the trajectory, we can also utilize the motion trajectories of any points in the start or end frame to guide the video frame interpolation process, even if the point may not be visible in the start or end frame.
>
> [1] Mochi 1: A new SOTA in open-source video generation models.
> [2] CogVideoX: Text-to-Video Diffusion Models with An Expert Transformer, Arxiv 2024.
>
> ---
>
> ### W#1: Although the method is effective, the novelty is limited.
>
> Thanks for pointing this out. We highlight two key technical contributions introduced in Framer. Firstly, we utilize **explicit long-term correspondences between start and end frames** to enhance video frame interpolation (VFI). The popular flow-based methods rely on short-term temporal correlations between adjacent frames, while the kernel-based and recent diffusion-based approaches implicitly learn these correlations from training data, neglecting the explicit relationships between input frames. Consequently, these methods struggle with input frames that exhibit significant differences. In contrast, our approach leverages long-term temporal guidance derived from user drag inputs or estimated point trajectories, resulting in improved quality and temporal consistency in the output videos.
> Secondly, we present **a novel point trajectory estimation method that integrates feature matching with a new bi-directional point tracking technique** to achieve accurate point trajectory guidance. Although a uni-directional point tracking method has been proposed in DragGAN, directly applying it to video frame interpolation leads to sub-optimal results, as demonstrated in Section 4.4 and Table S1. To tackle this, our motivation is that matched keypoints in both start and end frames offer distinctive features for effective point tracking. Based on this observation, we implement both forward and backward point tracking and introduce a bi-directional consistency check process to enhance tracking accuracy.
> Beyond these technical contributions, Framer introduces **a novel way of interaction in video frame interpolation to align with user intentions**. Users can unleash their creativity through simple and intuitive click-and-drag operations on input images. Based on these, **Framer supports novel downstream applications, such as image morphing and cartoon or sketch interpolation**, which require user customization and the ability to manage large differences between input frames.

---

> ### Author Response · Authors · 2024-11-22
> **Responses to Reviewer SBQc (2/2)**
>
> ---
>
> ### W#2:The motion and the difference between two key frames (first and last frames) are not complex.
>
> Thanks to your valuable comments. We add results on interpolating input frames with large differences in Fig. S14 in the Appendix. The results demonstrate the robustness of Framer. The associated videos have also been added to the Supplementary Material.
> In addition, we added [1] to the references in related work.
>
>
> [1] Training Weakly Supervised Video Frame Interpolation With Events, ICCV 2021.
>
> ---
>
> ### W#3: The image quality of some results shown in this paper degrades compared to the first input frame.
>
> Despite the advancements in video interpolation achieved by Framer over previous methods, we acknowledge that there is still potential for further improvement. To that end, we plan to enhance the video interpolation quality of Framer from several angles. **Firstly, the Variational Autoencoder (VAE) used in Stable Video Diffusion faces difficulties in reconstructing the details in video.** To tackle this, we intend to implement stronger VAEs that can improve the quality of video interpolation, such as CV-VAE [1] or other advanced VAE architectures. **Secondly, we currently train Framer using input videos with a resolution of 512x320.** In the future, we aim to process higher-resolution inputs and conduct training on an extensive dataset with more video data, which will contribute to an overall improvement in the generated video quality.
>
> [1] CV-VAE: A Compatible Video VAE for Latent Generative Video Models, NeurIPS 2024.
>
>
>
> ---
>
> ### W#4: The reconstruction metric of this method is not best when comparing it to related methods.
>
> Thank you for your valuable comments. We would like to emphasize that the current reconstruction metrics do not accurately reflect the quality of video generation. These metrics tend to assign low scores to other possible video interpolation results that align with real-world data distributions, while favoring videos with unnatural motion that merely match pixel distributions closely with the ground truth video. A similar discussion is made in VIDIM [1]. In the future, we will explore more reasonable quantitative metrics to reflect the quality of video frame interpolation.
> Furthermore, when Framer is guided by the trajectory of the ground truth video, denoted as "Framer with Co-Tracker (Ours)" in Tab. 1, it achieves the best results on all metrics, including the reconstruction metrics.
>
> [1]  Video Interpolation With Diffusion Models, CVPR 2024.

---

> ### Author Response · Authors · 2024-11-25
> **Welcome Further Feedbacks**
>
> Dear Reviewer **SBQc**,
> We would like to thank you for the thoughtful and constructive feedback. During the rebuttal, we performed additional experiments and analyses to address your concerns. The experiments and analyses in our response are summarized as follows.
> - We conduct experiments to show that Framer supports generating varying numbers of frames between input frames.
> - We discuss future directions to handle the scenarios when the number of corresponding points between frames is too few.
> - We add results on handling input frames with large motion and differences.
> - We clarify the novelty and contribution of this paper.
> - We discuss the reconstruction metric for quantitative comparison.
> - We discuss future directions for further improving image quality.
>
>
> We are wondering if the responses have addressed your concerns and would appreciate it if you considered raising your final rating. Looking forward to your further feedback!

---

> ### Author Response · Authors · 2024-11-27
> **Furthur Responses to Reviewer SBQc**
>
> Thanks for recognizing our efforts and thank you again for your valuable comments! We continue to address your other concerns below.
>
> ---
> To further investigate the reasons for image quality degradation, we conducted a series of analyses on the DAVIS-7 dataset.
> 1. **VAE Reconstruction Evaluation.** We began by assessing the reconstruction error of the Variational Autoencoder (VAE) by comparing the VAE-reconstructed video with the ground truth video. The results, summarized in the table below, indicate that the VAE achieved a PSNR of 27.84 and an SSIM of 0.8751, among other metrics. These findings suggest that while the VAE can effectively reconstruct the input video, some errors in detail remain.
>     |      |  PSNR   |  SSIM    |  LPIPS   |    FID   |  NIQE   |
>     |:----:|:-------:|:--------:|:--------:|:-------:|:--------:|
>     | VAE |  27.84 | 0.8751 | 0.1436  | 39.93  | 4.875  |
> 2. **Frame-by-Frame Quality Assessment.** Next, we evaluated the quality of each frame generated by the Framer independently. By design, the first and last frames of the Framer's output should align to the input start and end frames, respectively. This allows us to directly measure the Framer's capability to generate these boundary frames using standard reconstruction metrics. The results for each frame index are summarized in the table below. Our observations are as follows:
>     |      |  PSNR   |  SSIM    |  LPIPS   |    FID   |  NIQE   |
>     |:----:|:-------:|:--------:|:--------:|:-------:|:--------:|
>     | **Frame 0** |  27.55 | 0.8854 | 0.1331 | 37.26  | 4.943 |
>     | Frame 1 |  24.69 | 0.8418 | 0.1790 | 41.55 | 5.003 |
>     | Frame 2 | 23.11 | 0.8027 | 0.2124 | 43.67 | 5.012 |
>     | Frame 3 |   22.33 | 0.7804 | 0.2339 | 45.26 | 4.933   |
>     | **Frame 4** |  20.18   |  0.6850    |  0.2845   |    55.13   |  5.155   |
>     | Frame 5 |  22.01 | 0.7745 | 0.2409  | 52.36 | 5.027 |
>     | Frame 6 |  22.29 | 0.7832 | 0.2294  | 46.27 | 5.095 |
>     | Frame 7 |  23.17 | 0.8094 | 0.1998 | 43.26 | 5.068 |
>     | **Frame 8** | 27.85 | 0.8657 | 0.1227 | 40.26 | 4.973 |
>
>       - For the first (Frame 0) and last frames (Frame 8), the generated images closely resemble the ground truth images, showing comparable metrics to those of the VAE reconstruction. This indicates that image quality degradation at these points is primarily due to the reconstruction errors of the VAE.
>       - Moving from the first and last frames toward the middle of the sequence, there is a noticeable decline in the quality of the generated images. Particularly, Frame 4 exhibits the lowest quality among the generated frames. We attribute this decline to the fact that intermediate frames differ more significantly from input, demanding greater generative capability from the model. A similar trend is evident when comparing Tab. 1 and Tab. S3 in our manuscript, where different methods perform worse on the intermediate frames than on the average across all frames.
>
>
> In summary, we conclude that the primary cause of image quality degradation in the first and last frames is the insufficient reconstruction capability of the VAE. Conversely, the degradation observed in the middle frames is mainly due to the limited generation capability of the latent diffusion model. Further investigations are needed to deepen our understanding of the underlying factors resulting in image degradation.

---

> ### Author Response · Authors · 2024-11-28
> **Looking Forward to Further Discussions**
>
> Dear Reviewer **SBQc**,
>
> We sincerely appreciate your valuable efforts in reviewing our paper. As the Author-Reviewer discussion period is nearing its conclusion, we would like to kindly inquire if the reviewers have any additional questions or concerns regarding our previous responses. Please feel free to reach out if there are any remaining points that need clarification.
>
> Thank you!

---

### Official Review · Reviewer_urco · 2024-11-04

**Soundness:** 3
**Presentation:** 2
**Contribution:** 2
**Rating:** 6
**Confidence:** 3

**Summary:**

The paper proposes a new technique for generating frame interpolations that can incorporate user input to drive the interpolation process between two given frames. The method uses Stable Video Diffusion as the base model with the inputs modified to include both the first and last frame. Additionally, a control branch consisting of gaussian heatmaps is encoded and included in the model. The method is evaluated on a variety of image categories and on publicly available datasets.

**Strengths:**

The paper is mostly well written and easy to follow. The novelty here lies in the control branch for guiding the interpolation process that allows for fine-grained user-friendly control over the interpolation process.. The authors evaluate several existing methods on a variety of input types like real-world, sketch, cartoon, etc both qualitatively and quantitatively and produce high quality results.  Also included in a technique to produce the control signals automatically using bidirectional consistency checks. Several key design choices have been evaluated thoroughly, such as number of trajectories, bidirectional consistency.

**Weaknesses:**

- There is no mention of inference speed, training speed or overall training methodology (end-to-end, multi stage, pretraining, etc)
- There appears to be no underlying camera model for handling the "novel view synthesis" examples. Having such a model could potentially improve quality while providing another user-control.
- The paper isn't very easy to follow specifically around the trajectory update for autopilot mode section.

**Questions:**

- What are the differences in training / inference methodology between the autopilot and user-guided mode?
- Are there categories of videos where the results aren't as high quality as other categories?
- Could you explain a bit more about "considering that the conditions and the corresponding noisy latents are spatially aligned" near line 183?
- Can you explain how you transform 2D points into gaussian heatmaps? Are any other modalities like text used here? References to DragNUWA or DragAnything might be too recent. Consider also citing an older source such as from 2014 "Stacked Hourglass Networks for Human Pose Estimation" if it makes sense.
- Can the autopilot mode re-use the randomly sampled points ~line 197 instead of using feature matching to find new correspondences over time?
- Have you considered simpler alternatives for the Trajectory updating section?
- How do you handle camera motion in your framework?

---

> ### Author Response · Authors · 2024-11-22
> **Responses to Reviewer urco (1/3)**
>
> Thanks for your careful review and constructive suggestions. The questions are answered as follows.
>
> ---
>
> ### Q#1: What are the differences in training / inference methodology between the autopilot and user-guided mode?
>
> During training, both autopilot mode and user-interactive mode are trained using the same method. During inference, the key difference between the two modes is the source of the point trajectory guidance. For autopilot mode, the trajectories are initialized by connecting the matched keypoints in start and end frames, and updated using the proposed bi-directional point matching. Differently, for the user-interactive mode, the trajectories are simply obtained from the user interactions.
>
> ---
>
> ### Q#2: Are there categories of videos where the results aren't as high quality as other categories?
>
> Thank you for pointing this out. Here we examine potential scenarios where the model may underperform or encounter failures, particularly in cases where it fails to capture object semantics, or no suitable correspondences between the input frames can be found. The failure cases are presented in Fig. 13 in the updated manuscript and the accompanying videos are presented in the Supplementary Material.
>
> **(a) Fail to capture object semantics.**
> When a moving object appears blurred in the first and last frames of a sequence, it's sometimes difficult for the model to accurately interpret the object's semantics, which can lead to unrealistic generation results. As highlighted in DynamiCrafter [1], incorporating text guidance during video frame interpolation enhances the model's comprehension of moving objects and helps to resolve ambiguities associated with unclear objects. For this reason, we plan to introduce textual guidance to mitigate this problem in future work.
>
> **(b) Lack of suitable matching points between the first and last frames.**
> When there are no suitable matching points between the first and last frames, Framer struggles to utilize trajectory guidance effectively, resulting in sub-optimal video interpolation results. The model faces challenges when generating a scene where a character present in the initial frame exits while another character emerges from the background. As shown in Fig. 13, while Framer can produce reasonable video interpolation results, there is still noticeable distortion in the image. To address this problem, we are exploring the use of text guidance, as well as leveraging more advanced video models like Mochi [3] and CogVideo-X [4], to improve our handling of such scenes.
>
> We have added new sections to the paper to discuss the above issues, as shown in Sec. 4.5. In the future, we will use a stronger base model for video generation and introduce the guidance of text to achieve higher-quality video frame interpolation.
>
> [1] DynamiCrafter: Animating Open-domain Images with Video Diffusion Priors. ECCV 2024.
> [2] ToonCrafter: Generative Cartoon Interpolation. SIGGRAPH Asia 2024.
> [3] Mochi 1: A new SOTA in open-source video generation models.
> [4] CogVideoX: Text-to-Video Diffusion Models with An Expert Transformer, Arxiv 2024.
>
> ---
>
> ### Q#3: Could you explain a bit more about "considering that the conditions and the corresponding noisy latents are spatially aligned" near line 183?
> The noisy latent representation of the final frame ($z_t^n$) is derived by adding noise to the clean latent representation of the last frame ($z^n$). In other words, the feature at each pixel location of $z_t^n$ aligns with the feature present in the control condition $z^n$, once the Gaussian noise is removed. Consequently, these features are considered spatially aligned. To improve clarity, we updated the manuscript using the following description.
> "Additionally, we concatenate the latent feature of the last frame, $z^n$, with the noisy latent of the end frame, $z_t^n$, considering that $z_t^n$ is derived by adding noise to $z^n$."

---

> ### Author Response · Authors · 2024-11-22
> **Responses to Reviewer urco (2/3)**
>
> ---
>
> ### Q#4: Can you explain how you transform 2D points into gaussian heatmaps? Are any other modalities like text used here?
>
> Thanks for pointing this out. No text or other modalities are used in this context. We transform 2D points into Gaussian heatmaps, following the practice in Stacked Hourglass [1], DragNUWA [2], and DragAnthing [3]. Specifically, we initialize a canvas map with the same height and width of the input video, setting all values to zero. Subsequently, for each trajectory point at the coordinate position $p$, we create a grid region centered on this point with a pixel area of 41x41. The center of this area (coordinate $p$) is assigned a value of 1, while the values decrease in accordance with a Gaussian distribution as the distance from $p$ increases. The variance of this Gaussian distribution is set to 8 in both the horizontal and vertical directions. We added the implementation details to App. A of the updated manuscript.
>
> [1] Stacked Hourglass Networks for Human Pose Estimation. ECCV 2016.
> [2] DragNUWA: Fine-grained Control in Video Generation by Integrating Text, Image, and Trajectory. Arxiv 2024.
> [3] DragAnything: Motion Control for Anything using Entity Representation. ECCV 2024.
>
>
> ---
>
> ### Q#5: Can the autopilot mode re-use the randomly sampled points ~line 197 instead of using feature matching to find new correspondences over time?
>
> Thanks for your suggestion, but we can not re-use the points sampled during training.
> **During training**, since the ground truth video is available, we can obtain the matched keypoints on the end and middle frames by applying point tracking algorithms, such as Co-Tracker, after randomly sampling the keypoints on the first frame.
> **In contrast, during inference using autopilot mode, we only have the start and end frames, without access to the ground truth video.** Therefore, we need to obtain the correlation between keypoints on the first and last frames by performing feature matching on these two frames.
>
> ---
>
>
>
> ### Q#6: Have you considered simpler alternatives for the Trajectory updating section?
>
> Thanks for your good suggestion. We explored simpler trajectory updating strategies, such as omitting the bi-directional consistency check described in Sec. 3.3 and relying solely on the matching of the middle frame with the first frame for trajectory updates. However, these approaches resulted in a noticeable decline in performance, as demonstrated in Figure 12 and Table S1.
> Additionally, we also considered other trajectory updating methods, including the use of large language models (LLMs) to predict the coordinates of trajectory points, as outlined in the paper [1]. However, employing LLMs for trajectory prediction significantly increases inference overhead.
> It is important to emphasize that trajectory updating itself is relatively low in complexity. It primarily involves analyzing the neighborhood around the trajectory point and selecting the nearest neighbors based on similarity calculations. Furthermore, the time consumption associated with trajectory updating is negligible. Specifically, the trajectory updates only take 4.9% inference time during sampling.
>
> [1] LLM-grounded Video Diffusion Models, ICLR 2024.
>
>
>
> ---
>
> ### Q#7 & W2: How do you handle camera motion in your framework?
>
> Our method does not rely on explicit camera pose modeling for video frame interpolation, which brings additional benefits. Specifically, it reduces the demands on training data, as it does not necessitate camera pose annotations. This allows us to effectively leverage the massive amounts of video data available on the Internet. Experiments show that despite not explicitly modeling camera pose and simply training on open-world videos, our method still generalizes well to the interpolating frames of viewpoint-changing videos, as demonstrated in Fig. 7 and Fig. S9.
> In future work, we plan to introduce control modules for camera pose based on Framer pre-trained on open-world videos, similar to the approaches used in [1, 2]. This will enable accurate camera pose control and lead to higher-quality novel synthesis generation.
>
> [1] MotionCtrl: A Unified and Flexible Motion Controller for Video Generation, SIGGRAPH 2024.
> [2] CameraCtrl: Enabling Camera Control for Text-to-Video Generation, Arxiv 2024.

---

> ### Author Response · Authors · 2024-11-22
> **Responses to Reviewer urco (3/3)**
>
> ---
>
> ### W#1: There is no mention of inference speed, training speed or overall training methodology (end-to-end, multi stage, pretraining, etc)
>
> Thanks for your good suggestion. The training methodology is detailed in Sec. 4.1. The model is trained in two stages. Specifically, we first fine-tune the U-Net to accept the end frame conditions. Then, we train the controlling branch for point trajectory guidance.
> During the training of U-Net, we fixed the spatial attention and residual blocks, and only fine-tuned the input convolutional and temporal attention layers. The model is trained for 100K iterations using the AdamW optimizer with a learning rate of 1e-5.
> When training the control module, we fixed the U-Net and optimized the control module for 10K steps using the AdamW optimizer, with a learning rate of 1e-5. We obtained the point trajectories by pre-processing the video using the Co-Tracker.
> All training is performed on 16 NVIDIA A100 GPUs, and the total batch size is 16. The training takes about 2 days.
> During sampling, it takes about 4.64 seconds to generate 7 interpolated frames on the DAVIS-7 dataset. On average, it takes 0.66 seconds to produce a single interpolated frame.
>
>
> ---
>
> ### W#3: The paper isn't very easy to follow specifically around the trajectory update for autopilot mode section.
> We appreciate your feedback. In response to your concern, we include an algorithm to enhance the clarity of this section, as shown in Algorithm 1 in the Appendix.

---

> ### Author Response · Authors · 2024-11-25
> **Welcome Further Feedbacks**
>
> Dear Reviewer **urco**,
> We would like to thank you for the thoughtful and constructive feedback. During the rebuttal, we performed additional experiments and analyses to address your concerns. The experiments and analyses in our response are summarized as follows.
> - We discuss potential failure cases for Framer, including:
>   - Lack of suitable matching points between the first and last frames.
>   - Failure to capture object semantics.
> - We add Algorithm 1 in the Appendix to better illustrate the point trajectory estimation method.
> - We discuss alternatives for the trajectory updating.
> - We discuss camera motion control in Framer.
> - We clarify several aspects of Framer, including:
>   - The differences in training / inference methodology between the autopilot and user-guided mode.
>   - The "spatially alignment" near line 183.
>   - More training and inference details.
>
>
> We are wondering if the responses have addressed your concerns and would appreciate it if you considered raising your final rating. Looking forward to your further feedback!

---

> ### Author Response · Authors · 2024-11-28
> **Looking Forward to Further Discussions**
>
> Dear Reviewer **urco**,
>
> We sincerely appreciate your valuable efforts in reviewing our paper. As the Author-Reviewer discussion period is nearing its conclusion, we would like to kindly inquire if the reviewers have any additional questions or concerns regarding our previous responses. Please feel free to reach out if there are any remaining points that need clarification.
>
> Thank you!

---

> > ### Comment · Reviewer_urco · 2024-12-03
> >
> > Thank you, authors, for the thorough rebuttal! You've addressed most of my questions and I've updated my review to reflect this.

---

> > > ### Author Response · Authors · 2024-12-03
> > >
> > > Thank you for your feedback! We’re glad to hear that our response has resolved your concerns. We also appreciate your insightful reviews, which have greatly improved the quality of our paper.

---

### Author Response · Authors · 2024-11-25
**Summary of Responses to Reviewers**

We thank all reviewers for their valuable feedback, and are encouraged that the reviewers appreciate our clear motivation (**hifJ**), novelty (**urco**), detailed evaluation (**urco**, **SBQc**, **hifJ**), and impressive performance (**urco**, **SBQc**, **4487**, **hifJ**). Here, we provide a high-level summary of the changes we've made to address your concerns, and conclude with an overview of our key contributions.

---
Here is the summary of updates that we've made to address the reviewers' concerns:
- We discuss failure cases under the following scenarios, and discuss potential improvements. (**urco**, **hifJ**)
  - Lack of suitable matching points between the first and last frames.
  - Failure to capture object semantics.
- We clarify the novelty and contribution of this paper. (**SBQc**, **4487**)
- We evaluate and discuss the inference speed of Framer. (**urco**, **4487**)
- We conduct experiments to demonstrate the robustness and generalization capability of Framer, including
  - Generating varying numbers of frames between the start and end inputs. (**SBQc**)
  - Handling large motion and differences between input frames. (**SBQc**)
  - Effectiveness under complex trajectories, measured by video quality and trajectory accuracy. (**4487**)
  - Effectiveness under low-resolution inputs. (**4487**)
  - Benchmarking results on Vimeo90K, X4K1000FPS, and Adobe240, and evaluation with NIQE metric. (**4487**)
  - Benchmarking results on the task of cartoon interpolation and sketch interpolation. (**4487**)
- We improve paper writing from the following aspects.
  - We discuss previous work on tackling motion ambiguity and interactive frame interpolation. (**4487**)
  - We discuss the applications of video frame interpolation methods. (**4487**)
  - We add an Algorithm to better illustrate the point trajectory estimation method. (**urco**)
  - We add more details on transforming 2D points into Gaussian heatmaps. (**urco**)
  - We discuss camera motion control in Framer. (**urco**)

---
The contribution of our work is summarized as follows.
- Framer introduces **a novel way of interaction in video frame interpolation to align with user intentions**. Users can unleash their creativity through simple and intuitive click-and-drag operations on input images.
- In Framer, we utilize **explicit long-term correspondences between start and end frames** to enhance video frame interpolation (VFI). Previous methods struggle with input frames that exhibit significant differences. In contrast, our approach leverages **long-term temporal guidance derived from user drag inputs or estimated point trajectories**, resulting in improved quality and temporal consistency in the output videos.
- we present **a novel point trajectory estimation method that integrates feature matching with a new bi-directional point tracking technique** to achieve accurate point trajectory guidance. Although a uni-directional point tracking method has been proposed in DragGAN, directly applying it to video frame interpolation leads to sub-optimal results, as demonstrated in Section 4.4 and Table S1. To tackle this, our motivation is that matched keypoints in both start and end frames offer distinctive features for effective point tracking. Based on this observation, we implement both forward and backward point tracking and introduce a bi-directional consistency check process to enhance tracking accuracy.
- **Framer supports novel downstream applications, such as image morphing and cartoon or sketch interpolation**, which require user customization and the ability to manage large differences between input frames.

---

### Meta-Review · Area_Chair_fWN3 · 2024-12-21

**Metareview:**

The paper introduces Framer, an method for using user-provided trajectories as guidance in frame interpolation or morphing. Reviewers praised the paper's clear motivation and impressive performance, but some questioned its practical applicability and technical innovation.  Reviews were unanimously positive, and thus the paper has been chosen for acceptance.

**Additional Comments On Reviewer Discussion:**

Reviews began positive, and improved over the discussion period due to an extensive set of additional experiments and details provided by the authors.

As they remain, all reviewers recommend acceptance of the paper.

---

### Decision · Program_Chairs · 2025-01-22

Accept (Poster)